# Microphysics of Arctic Stratiform Boundary-layer Clouds during ARCSIX

Alexei V. Korolev[1] and R. Paul Lawson[2]

[1]Environment and Climate Change Canada
4905 Dufferin Street
Toronto, ON M3H 5T4
Canada

[2]SPEC Incorporated
3022 Sterling Circle
Suite 200
Boulder, Colorado, 80301
USA

*Correspondence to*: R. Paul Lawson (plawson@specinc.com)

**Abstract.** Clouds have a major impact on rapidly decreasing sea ice in the Arctic, yet much is still unknown how cloud microphysics influences cloud development. In situ and remote data were collected by the NASA P-3 and SPEC Inc. Learjet research aircraft in Arctic stratiform boundary-layer clouds over the oceans and sea ice bordering northern Greenland between 25 May and 15 August 2024 during the ARCSIX project. Both aircraft carried a suite of nearly identical state-of-the-art microphysical sensors. Additionally, the P-3 was equipped with aerosol and remote-sensing instrumentation and the Learjet was equipped with a zenith/nadir Ka-band radar. The length of clouds examined remotely and in-situ by the two aircraft totaled 12,417 km, with 6,966 km of in-situ measurements. Mixed-phase clouds were sampled during 60.5% of time in cloud, and all-liquid clouds were measured 39.5% of the time. Cloud-top temperatures were ≥ - 9 °C during 90% of the stratiform boundary-layer cloud investigations. Single-layer mixed-phase clouds sampled with cloud-top temperatures ≥ - 4 °C often contained concentrations of ice particles more than five orders of magnitude higher than measured concentrations of ice-nucleating particles. Despite the high ice concentrations, microphysical conditions supporting secondary ice production were seldom present. In contrast, in some clouds where environmental conditions met commonly accepted criteria for secondary ice production, ice particle concentrations were closer to what is expected from primary nucleation. The quality of measurements was unprecedented, but results from our preliminary analysis raise more questions about primary and secondary nucleation mechanisms than they answer.

# 1 Introduction

The International Panel on Climate Change (IPCC) has consistently issued Assessment Reports concluding that aerosols and clouds are two of most significant contributors to the rate of warming in the Arctic (IPCC, 1990, 1996, 2001, 2007, 2013, 2023), which is now shown to be more than twice that of the global average (IPCC, 2023). The NASA Arctic Radiation-Cloud-Aerosol-Surface Interaction Experiment (ARCSIX) was designed to increase our knowledge of how aerosols, clouds and surface properties contribute to warming temperatures and the melting of sea ice in the Arctic.

Arctic sea ice extent has declined by more than 40% since 1979 (Meier et al., 2017; Meredith et al., 2019) and sea ice thickness by ~70% since the early 1980s (Schweiger et al., 2011). As sea ice lessens open ocean water increases, causing greater absorption of solar radiation, increased warming and low-level (i.e., boundary-layer) cloud cover (Kay and Gettleman, 2009; Alkama et al., 2020). Arctic stratiform boundary-layer clouds (SBCs) provide positive feedback whereby penetration of solar radiation increases melting and upwelling thermal radiation is trapped by the overlaying cloud layer (Tsay et al., 1989; Curry and Ebert, 1992; Schweiger and Key, 1994; Walsh and Chapman, 1998; Intrieri et al., 2002; Sandvik et al., 2007; Serreze and Barry, 2011; Morrison et al., 2012). Climate model predictions suggest that the Arctic Ocean will become ice-free sometime between 2030 and 2050 (Jahn et al., 2016).

The Greenland Ice Sheet (GIS) is by far the largest orographic feature in the Arctic and melting of the GIS would have a significant global impact. The GIS covers 82% of the area of Greenland with an average elevation of 1,500 m and a maximum height of 3,255 m (Bamber et al., 2001). The high topography strongly enhances Northern Hemisphere meridional heat exchange (Kristjánsson and McInnes, 1999) and influences the location of the Icelandic

Low. Melting of the GIS could weaken the thermohaline circulation, which transports warm, saline surface water poleward with a deep, overturning return flow of cold, less saline water. The overturning takes place in the Greenland, Irminger and Labrador Seas (Broecker et al., 1990). Much of Western Europe benefits from this heat flux into the high latitudes and significant cooling can be expected with a weakened thermohaline circulation. Melting of the GIS would increase sea level by about 6 m and have a devastating effect on coastal areas (IPCC, 2023).

The first airborne investigations of stratus clouds in the Arctic were conducted by the Russian Arctic and Antarctic Research Institute in the late 1950's (Dergach et al.,1960; Koptev and Voskresenskii, 1962). More recently, SBCs have been the focus of several airborne investigations since the early 1980's (Herman and Curry, 1984; Curry, 1986; Curry and Ebert, 1992; Hobbs and Rangno, 1998; Curry et al., 2000; Lawson et al., 2001; Verlinde et al., 2007; Gayet et al., 2009; Lawson and Zuidema, 2009; McFarquhar et al., 2011; Mioche et al., 2017; Wendisch et al., 2017;

Järvinen et al., 2023). Curry (1986) analyzed data collected by the National Center for Atmospheric Research (NCAR) Electra research aircraft during the Arctic Stratus Experiment (ASE) over the Beaufort Sea in June 1980. She found that SBCs often existed in multiple layers, displayed spatial inhomogeneity and contained drizzle. Using ASE data and a numerical model, Curry and Ebert, (1992) determined that mixed-phase clouds in the Arctic are abundant. In a review article, Morrison et al., (2012) cites long-term, ground-based observations from Shupe et al., (2011) showing

that mixed-phase clouds cover large swaths of the Arctic throughout the year. Shupe et al., (2011) and Morrison et al., (2012) further explain that the high frequency of occurrence of Arctic mixed-phase clouds is largely owing to their longevity.

Several aircraft investigations in the Arctic report observations of mixed-phase clouds containing concentrations of ice crystals greater than what would be expected from primary nucleation at the coldest cloud

temperature. Curry et al., (2000) observed SBCs over the Beauford Sea where mixed-phase cloud was sampled within a temperature range of -4° to -6 °C in the presence of drizzle, but not in colder clouds (-12 °C) when only small cloud drops were present. Gayet et al., (2009) found that all-liquid SBCs with drizzle existed with a cloud-top temperature -4° C in the warm-section of a cold front, and that mixed-phase clouds without drizzle occurred in the cold sector with cloud-top temperatures of -6 °C. Wendish et al., (2017) documented mixed-phase conditions in SBCs north of Svalbard

within the temperature range of -3° to -7 °C. Mioche et al., (2017) report results from four international airborne campaigns that investigated single-layer SBCs staged in the European Arctic region. They noted that all-liquid cloud layers often existed above mixed-phase layers. Järvinen et al., (2023) analyzed data from six aircraft case studies collected during the ACLOUD (Arctic CLoud Observations Using airborne measurements during polar Day) campaign. They also found that all-liquid clouds were observed over mixed-phase layers with cloud-top temperatures

as warm as -3.8 °C. Lawson et al., (2011) show data collected on 29 May 2008 from a cloud particle imager (CPI) and 4-pi radiometer installed on a tethered balloon at Ny-Ålesund, Svalbard. Their Figure 4 shows a vertical profile of CPI images with a region of water drops from 550 to 800 m (-3° to -4.5 °C), and a region below from 550 m to 200 m (-3° to -1 °C) with ice particles. This regime of water-over-ice persisted for two hours while the balloon was lowered and raised through cloud, providing true vertical profiles through the cloud while cloud particles were advected

horizontally. After two hours of vertical profiling the balloon was raised to 810 m where it exited cloud top. The 4-pi radiometer attached to the balloon indicated that clear sky existed above cloud top.

Also, it should be noted that measurements of relatively high ice concentrations in stratiform clouds are not unique to the Arctic. For example, Mossop et al., (1968) found ice concentrations three orders of magnitude greater than ice nucleating particle (INP) measurements in a cumulus cloud at -4 °C over the Southern Ocean. Yang et al. (2020) found ice concentrations much higher than expected from primary nucleation at -8 °C in tropical stratiform clouds.

## 2    The ARCSIX Project: Goals, Flight Profiles, Instrumentation and Dataset Overview

The ARCSIX campaign was staged from the Pituffik Space Base, Greenland (formerly Thule Air Force Base) in the Spring (25 May – 13 June) and Summer (25 July – 15 August) of 2024. The overarching goal of ARCSIX is to quantify the contributions of surface properties, clouds, aerosol particles, and precipitation to the Arctic summer surface radiation budget and sea ice melt. Three research aircraft (Fig. 1) collected in-situ and remote observations: The NASA P-3 was equipped with a suite of in-situ cloud particle probes and aerosol sensors, as well as several remote sensors. The SPEC Learjet was equipped with cloud particle sensors and a zenith/nadir Ka-Band radar. The NASA GIII provided high-altitude remote observations. In this paper we use a case-study approach to analyze microphysical data collected by the P-3 and Learjet.

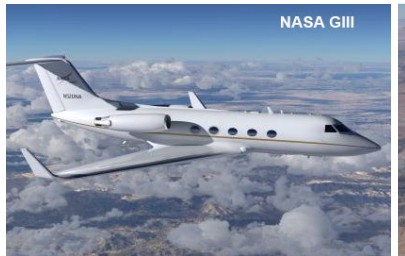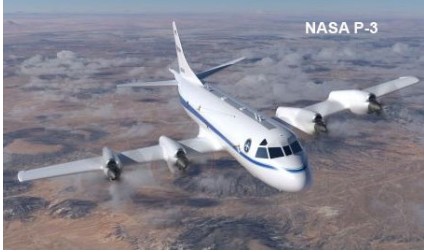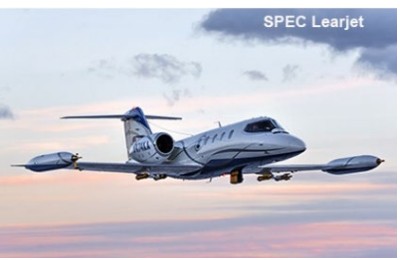

**Figure 1**. Three research aircraft deployed to Pituffik Space Port for ARCSIX. Photo Credits – NASA GIII and P-3: NASA Langley Research Center; SPEC Learjet: Code 10 Photography.

ARCSIX in-situ data were collected by the P-3 during 19 flight missions in the spring (30 May – 13 June 2024) and summer (24 July – 16 August 2024), and 11 missions by the Learjet during the summer deployment. The total length of clouds examined remotely and in situ by the two aircraft totaled 12,417 km, with 6,966 km in situ measurements. Mixed-phase clouds were sampled during 60.5% of time in cloud, and all-liquid clouds were sampled during 39.5% of the time. These percentages agree with Shupe et al., (2011), who found that when summer Arctic boundary-layer clouds are present, they are mixed-phase 60% of the time and all liquid 40% of the time. The Shupe et al., (2011) dataset was averaged over three Arctic stations (Barrow, Alaska; Eureka, Canada; and from a ship over the Beaufort Sea). Our emphasis in this paper is on clouds comprised of both single and multi-layers with cloud-top temperature ($T$) warmer than - 9° C, which constitutes over 90% of the dataset (excluding cirrus encounters during transit flights), although a few deeper systems with colder cloud top temperatures were also sampled.

Figure 2a shows the P-3 and Learjet flight tracks where the aircraft sampled SBCs with cloud-top $T$ > -9° C, and Fig. 2b shows flight tracks as a function of surface conditions. The main flight pattern for cloud in-situ microphysical and remote radiation studies is depicted in Fig. 3. In this scenario two or three aircraft are stacked vertically and either the P-3 or Learjet makes porpoising maneuvers from just below cloud base to sample precipitation (if any) and to just above cloud top to depict temperature behavior up to the inversion level. The ARCSIX science

team dubbed this a "Wall Pattern". When the Learjet was available it was the primary in-situ aircraft since the KPR
135 recorded up/down measurements and the P-3 had much larger complements of remote sensing and aerosol
instrumentation.

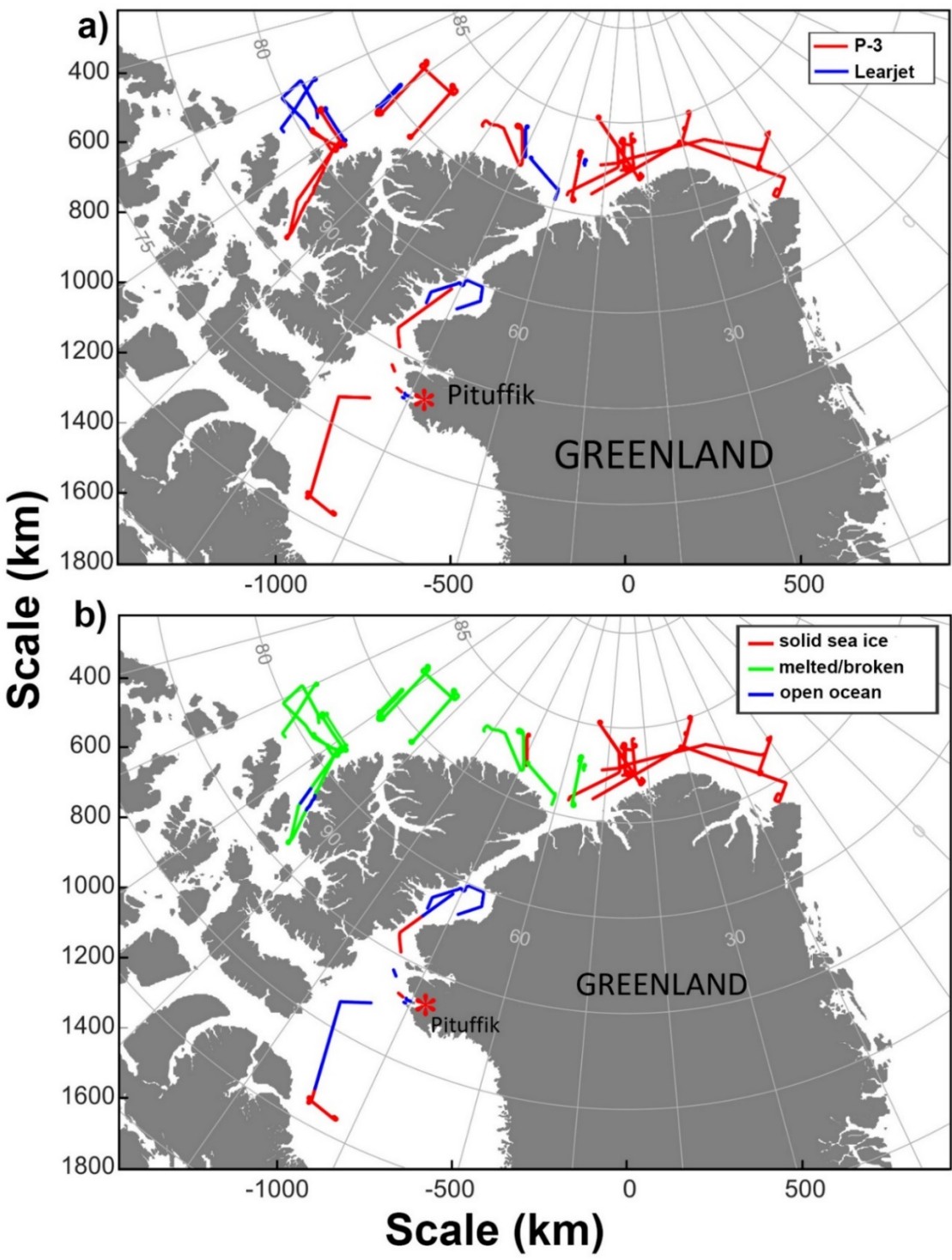

**Figure 2**. a) Flight tracks from the P-3 and Learjet in SBCs with cloud-top temperature > -9°
C, and b) P-3 and Learjet flight tracks from a) coded to show underlying surface conditions.

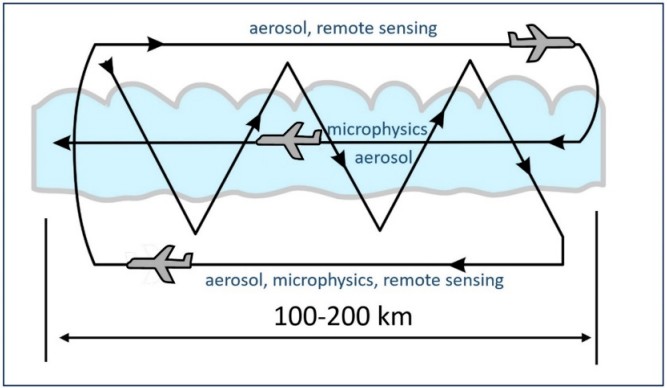

**Figure 3**. Example of flight patterns for investigating SBCs during ARCSIX.

ARCSIX data were collected with state-of-the-art in-situ and remote sensors. Table 1 shows a list of cloud probes, their measurement range and type of measurement. Three probes that had not previously been flown on the P-3 and Learjet were introduced in this project. The 2D-Gray probe has 10-μm pixel resolution and produces gray images at more than an order of magnitude greater than the data rate of previous gray probes. The HVPS-4 incorporates four independent probes, which provide two overlapping orthogonal views with 50-μm pixel resolution, and two overlapping orthogonal views with 150-μm pixel resolution. The horizontal and vertical views of the same particle show the deformation of falling raindrops, and also provide data for reconstructing a 3D view of ice particles. The phase particle spectrometer (PPS) probe flew on each aircraft and were prototypes that were developed just prior to the field campaign. The PPS contains a CPI with 0.7 μm pixel resolution and a 2D-Gray probe with 5-μm pixel resolution. While both PPS prototype instruments functioned during the project, the data are considered preliminary and need to be used with consultation from the manufacturer (SPEC Inc.).

The P-3 flew an SEA WCM-2000 LWC/TWC (total water content) probe. The WCM has two heated cylinders, 0.5 and 2-mm diameter, for measuring LWC and a heated scoop approximately 4-mm in diameter for measuring TWC (Lillie et al., 2005). The Learjet flew a Nevzorov LWC/TWC probe with 2-mm and 3-mm heated cylinders, and an 8-mm heated inverted cone (Korolev et al., 1998). The KPR is a Ka-band precipitation radar developed by ProSensing, Inc, (Pazmany and Haimov, 2018). It extended in front of the wing of the Learjet and switched from zenith to nadir views at 5 Hz with with a vertical resolution of 15 meters. The radar produced interleafed short radio frequency (RF) and linear frequency modulation (FM) pulses. During processing, a five-point, two-dimensional median filter was applied to smooth the measurements and remove clear-sky noise. The radar has a "deadband" that extends 150 m in both zenith and nadir directions from the aircraft, but the porpoising maneuvers provide complete cloud coverage when the aircraft descended/ascended in a slant direction. The KPR is insensitive to low concentrations of small cloud droplets (i.e., $< \sim 30$ μm diameter), but registers returns from drizzle and ice particles in sufficient concentrations that are larger than about 100 μm.

Table S1 and Table S2 show detailed lists of cloud properties for the P-3 and Learjet, respectively, including cloud base and cloud top heights and temperatures, maximum cloud-drop and ice-particle concentrations, maximum drop and ice particle sizes, presence of drizzle, maximum liquid water content (LWC), ice particle habits, thermodynamic phase, cloud top inversion, underlying surface and comments. Figure 4 shows normalized frequency

diagrams of mean values of cloud drop and ice particle parameters in mixed-phase clouds for the P-3 (red), Learjet (yellow) and both aircraft (blue). The data in Fig. 4 were collected in mixed-phase clouds with $T \geq$ -9 °C, drop concentration $\geq 3$ cm$^{-3}$. ice particles > 150 μm in concentrations $\geq 0.04$ L$^{-1}$, and altitude < 3 km. For the plots in Fig. 4, drop concentration, LWC and effective drop diameter (D$_{eff}$) were computed from FCDP measurements; ice particle concentration, IWC and D$_{max}$ were computed from 2D-Gray particle images with max particle size > 150 μm.

Figures 4a, b show that there was good agreement between P-3 and Learjet measurements of LWC and IWC. Figure 4a shows that about 90% of the mixed-phase clouds had LWC $\leq 0.3$ g m$^{-3}$, while Fig. 4b shows that about 90% of mixed-phase clouds had IWC $\leq 0.05$ g m$^{-3}$. Much larger maximum IWCs (0.4 to 0.5 g m$^{-3}$) were only observed in aggregates of columns and needles precipitating below some cloud bases. Figure 4c shows that approximately 70% of liquid and mixed-phase clouds contain a drop concentration less than 50 cm$^{-3}$ and over 90% less than 100 cm$^{-3}$. This is indicative of low aerosol loading, which is typical for the Arctic environment. Low droplet concentrations in clouds generally result in broad drop-size distributions due to the lower competition for water vapor, which can result in formation of drizzle (drop diameter $\geq 50$ μm). Drizzle in ARCSIX SBC's did not exceed ~300 μm in diameter. The low drop concentration and broad drop size distribution also results in a D$_{eff}$ = 22 to 24 μm (Fig. 4d), which is larger than typically found in mid-latitude continental clouds (Pruppacher and Klett 2010).

Figure 4e shows that about 50% of measured particle size distributions had a maximum ice dimension $\leq 500$ μm. However, much larger (1 to 3 mm) columnar ice particles were often found near the bottoms of SBCs, and in clusters of needles, sheaths and columns up to 1 cm were observed precipitating below some cloud bases. Figure 4f shows that about 90% of ice particles were found in concentrations $\leq 5$ L$^{-1}$. Curiously, as shown in Fig. 5, the highest mean ice particle concentrations were observed in the temperature range of -2 to -4 °C. The relatively high ice concentrations within this warm temperature range, where the concentration of INPs was undetectable, have also been reported in some previous Arctic campaigns (see Introduction). The trend of mean ice concentration versus temperature for the spring, summer, and combined deployments in Fig. 5 is opposite of what is expected from primary nucleation (Pruppacher and Klett 2010). One would normally expect the higher ice concentration at the warmer temperatures in Fig. 5 to be attributed to a secondary ice process (SIP). However, the relatively high ice concentrations were mostly measured in regions where environmental conditions for SIP were not observed. We go into this in more detail in Section 4.2.1.

As shown in Figures 4 and 5, mean ARCSIX ice particle concentrations in mixed-phased SBCs at $T >$ -6° C were typically low, ranging from about 1 to 5 L$^{-1}$ with notable exceptions that ranged from 10 to 50 L$^{-1}$, which exceeded INP measurements by several orders of magnitude in ARCSIX clouds (Perkins, 2025 – Personal Communication). Based on a close examination of CPI and 2D-Gray images, ice particle sizes varied from 15 μm in mixed-phase cloud with freshly nucleated ice, to 1-cm aggregates formed from clusters of needles, sheaths and columns below cloud base. Mioche et al., (2017) and Järvinen et al., (2023) report smaller maximum ice particle sizes and maximum values of IWC in Arctic clouds. This may be due to a combination of factors, including the coarser resolution of the Cloud Imaging Probe (CIP), the lack of reported large clusters of columnar ice, and different software algorithms for processing IWC. They also attributed relatively high ice concentrations to rime-splintering. We show in Section 4 that

high ice concentrations were sometimes observed in ARCSIX clouds where environmental conditions for rime-splintering did not exist.

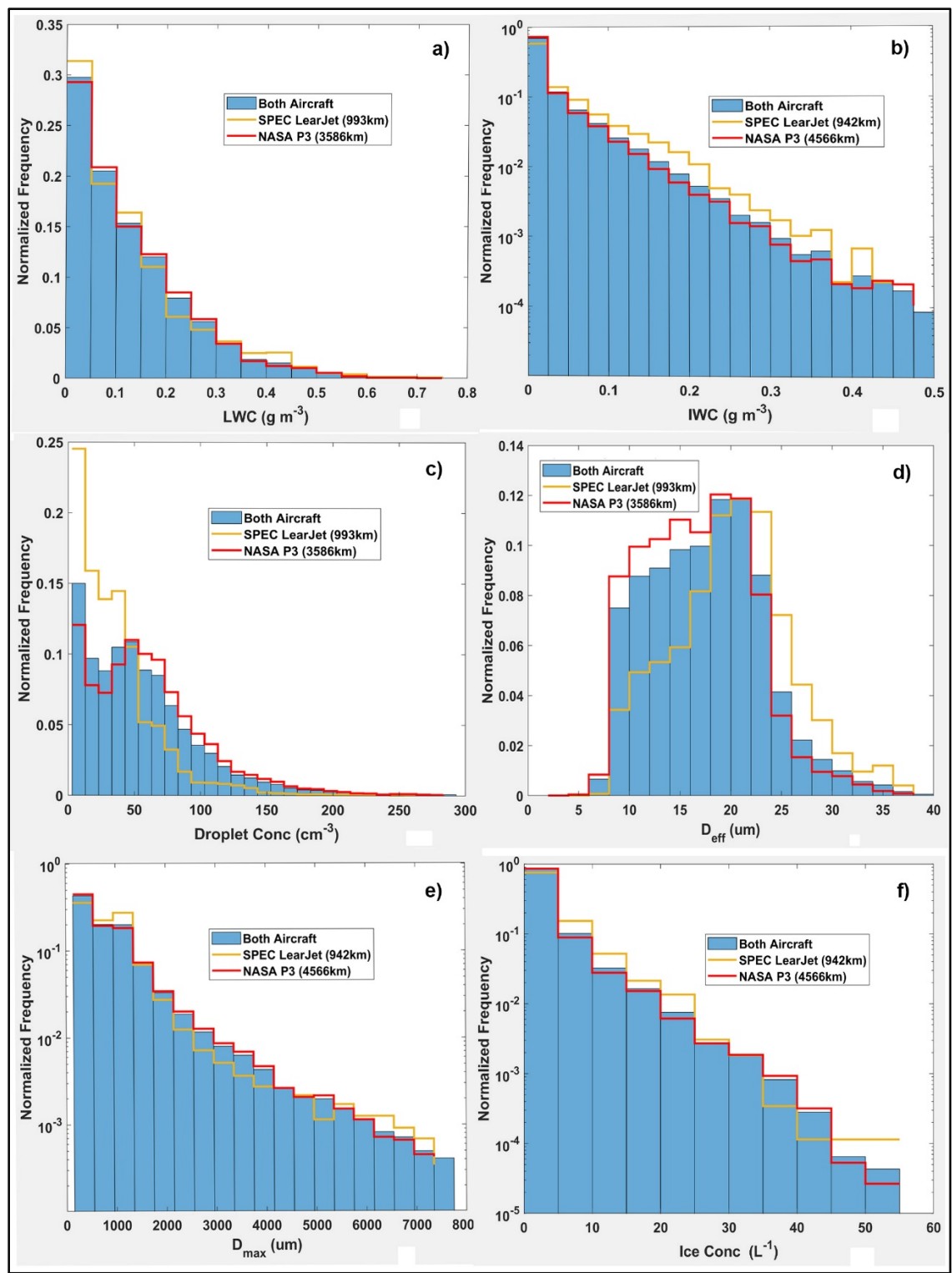

**Figure 4**. Normalized frequency histograms of mean data collected in Arctic mixed-phase SBCs with measurements from the Learjet (yellow), P-3 (red) and combined P-3 and Learjet (blue): a) LWC, b) IWC, c) drop concentration, d) maximum effective ice particle size ($D_{max}$), and effective drop diameter ($D_{eff}$).


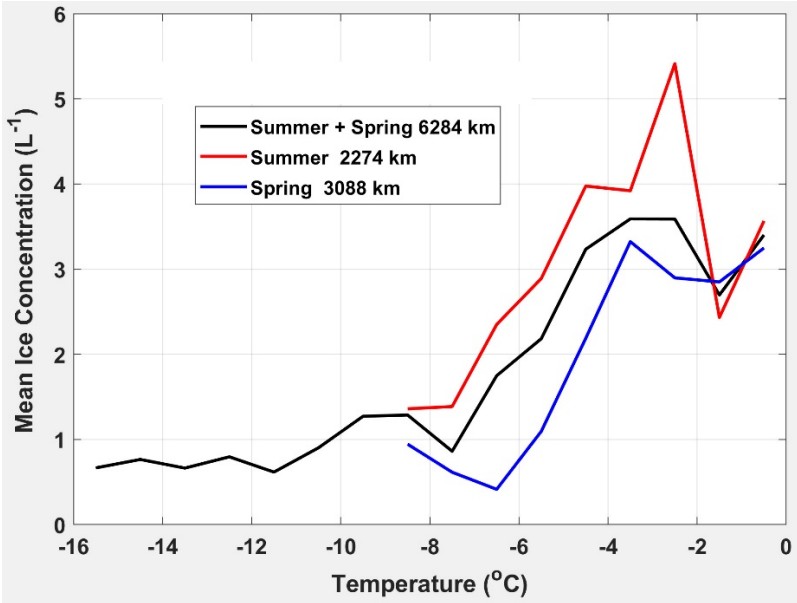

**Figure 5.** Plot of mean ice concentration in mixed-phase clouds as a function of temperature from P-3 and Lear data during the summer deployment (red), the P-3 during the spring deployment (blue), and summer and spring deployments combined (black).

In previous airborne projects, it was not always possible to rigorously determine if ice particles from colder clouds aloft seeded warmer SBCs below. In results presented in this paper, we have made careful examinations of lidar data, clear-air measurements from cloud particle probes, and observations of higher clouds from video cameras installed on the aircraft to rule-out potential seeding from aloft. We explore possible reasons for the anomalous observations of high ice concentrations at $T \geq -4°C$, but definitive explanations based on our current understanding of primary nucleation and SIP, are lacking.

**Table 1.** Lists of microphysical instrumentation installed on the NASA P-3 and SPEC Learjet.

| NASA P3 | | | | | SPEC LearJet | | | |
|---|---|---|---|---|---|---|---|---|
| **Probe Name** | **Sensor** | **Measurement Range** | **Type** | | **Probe Name** | **Sensor** | **Measurement Range** | **Type** |
| Hawkeye | 2DS-H | 50 - 6400 μm | OAP (1bit) | | 2DS | 2DS-H | 10 - 1280 μm | OAP (1bit) |
| | 2DS- | 10 - 1280 μm | OAP (1bit) | | | 2DS-V | 10 - 1280 μm | OAP (1bit) |
| | Fast CDP | 1.5 - 50 μm | scattering | | Fast CDP | Fast CDP | 1.5 - 50 mm | scattering |
| | CPI | 1024x1280 @ 2.3 μm | imaging | | 2DSGray | 2DSGray | 10 -1280 mm | OAP (2bit) |
| 2DSGray | 2DSGray | 10 -1280 μm | OAP (2bit) | | PPS | PPS-CPI | 877x2600 @ 0.7 μm | imaging |
| PPS | PPS-CPI | 877x2600 @ 0.7 μm | imaging | | | PPS-2DSGray | 5 -640 μm | OAP (2bit) |
| | 2DSGray | 5 -640 μm | OAP (2bit) | | Fast FSSP | Fast FSSP | 1.5 - 50 μm | scattering |
| Fast 2DS | Fast 2DS-H | 10 - 1280 μm | OAP (1bit) | | HVPS-4 | HVPS4-H | 50 - 6400 μm | OAP (1bit) |
| | Fast 2DS-V | 10 - 1280 μm | OAP (1bit) | | | HVPS4-V | 50 - 6400 μm | OAP (1bit) |
| | Fast CDP | 1.5 - 50 μm | scattering | | | HVPS4-H | 150 - 19200 μm | OAP (1bit) |
| HVPS-4 | HVPS4-H | 50 - 6400 μm | OAP (1bit) | | | HVPS4-V | 150 - 19200 μm | OAP (1bit) |
| | HVPS4-V | 50 - 6400 μm | OAP (1bit) | | Nevzorov probe | LWC1 | 0.01 - 3 g/m3 | hot wire |
| | HVPS4-H | 150 - 19200 μm | OAP (1bit) | | | TWC2 | 0.01 - 3 g/m3 | hot wire |
| | HVPS4-V | 150 - 19200 μm | OAP (1bit) | | | LWC3 | 0.01 - 3 g/m3 | hot wire |
| WCM | TWC | 0.01 - 3 g/m3 | hot wire | | Rosemount icing detector | RICE | > 0.01 g/m3 | icing detector |
| | LWC1 | 0.01 - 3 g/m3 | hot wire | | KPR | Ka-band Radar | >-10 dBz | Precip Radar |
| | LWC2 | 0.01 - 3 g/m3 | hot wire | | | | | |
| Rosemount icing detector | RICE | > 0.01 g/m3 | icing detector | | | | | |

## 3    Cloud Structure

Despite the seemingly visual homogeneity of the appearance cloud top observed from the forward-looking cameras on the P-3 and Learjet, the in-situ observations showed high spatial variability of cloud microphysical parameters. Cloud droplet and ice particle concentrations, size distributions, LWC, IWC and extinction coefficient varied over spatial scales extending from hundred of meters to tens of kilometers. Figure 6 shows an example of a single-layer SBC with high spatial intermittency of thermodynamic phase, where liquid cloud segments are adjacent

to mixed-phase segments. The measurements were performed over open water in the Baffin Bay by the P-3 on 10 June 2024 (RF08). Figure 6a shows that the aircraft was in level flight at 370 m (-3.3° C). Cloud top was measured at 430 m with $T$ = -4° C, and cloud base at 255 m (-2.0° C). This layer was topped by a positive temperature inversion with maximum temperature $T$ = +2° C, which excluded seeding of this cloud layer by ice particles. Figures 6 b – d show b) inhomogeneity in measurements from the WCM-2000 LWC sensors; c) ice particle concentrations determined for

particles with diameter > 150 μm from the fast 2D-S, 2D-Gray and 50-μm channel of the HVPS-4 probes; d) and maximum particle diameters from the fast 2D-S and 2D-Gray probes. The blue-shaded region in Fig. 6 highlights an example of an all-liquid section of cloud, and the yellow-shading shows an example of a mixed-phase region. While LWC varies from about 0.1 to 0.25 g m$^{-3}$ in the mixed-phase region (Fig. 6b), the average IWC shown in Fig. 6d is an order of magnitude less than LWC. IWC was typically at least an order of magnitude less than LWC in mixed-phase

clouds, as shown earlier in Fig. 4c,d, here in Fig. 6b, d, and later in Figs 10 and 12. However, IWC in aggregates of columns and needles precipitating below cloud base often equaled or exceeded LWC higher in the cloud.

         Examples of particle images from the 2D-Gray and Hawkeye CPI probes are shown below the time series in Fig. 6. In this example, there are no drizzle-size drops and the recognizable ice habits are needles, sheaths and columns. The inhomogeneity in microphysics in this single-layer cloud is intermittent without any periodic structure that would

suggest wave dynamics, or from isolated sources of INPs at the sea surface (e.g. ship emissions). On the other hand, some SBCs displayed more of a cellular structure of ice development. Figure 7 shows an example of a SBC with cellular structure observed during the summer deployment by the Learjet on 30 July 2024 (RF03). In this case, the Learjet is flying below cloud base (395 m -5.8° C) at 190 m (-3.7° C) with a cloud top of 650 m (-7.0° C). The up/down Ka-band radar provides 1-Hz averaged measurements starting 150 m from the aircraft. The cellular structure in the

radar data (Fig. 7b) is at a spatial scale from about 200 to 500 m and correlates well with measurements of ice water content (IWC) in Fig. 7c.

         Examples of particle images from the 2D-Gray, HVPS4 50- and 150-μm channels in a region with relatively high IWC are shown below the time series in Fig. 7. As expected in this temperature range, the predominate ice habits are again needles, sheaths and columns, with several smaller irregular-shaped ice particles seen in the CPI images. In

this regime with high aspect-ratio ice crystals, IWC is best estimated using particle area as the operator rather than maximum particle dimension (Baker and Lawson, 2006; Lawson and Baker, 2006). While, needles, sheaths and columns grown by vapor diffusion are expected in a mixed-phase cloud with a cloud-top temperature of -7.0° C, the irregular ice particles are not. The origin of ice at these warm temperatures, and the irregular ice shapes, are difficult to explain. We discuss these subjects in more detail in Sections 4 and 5.


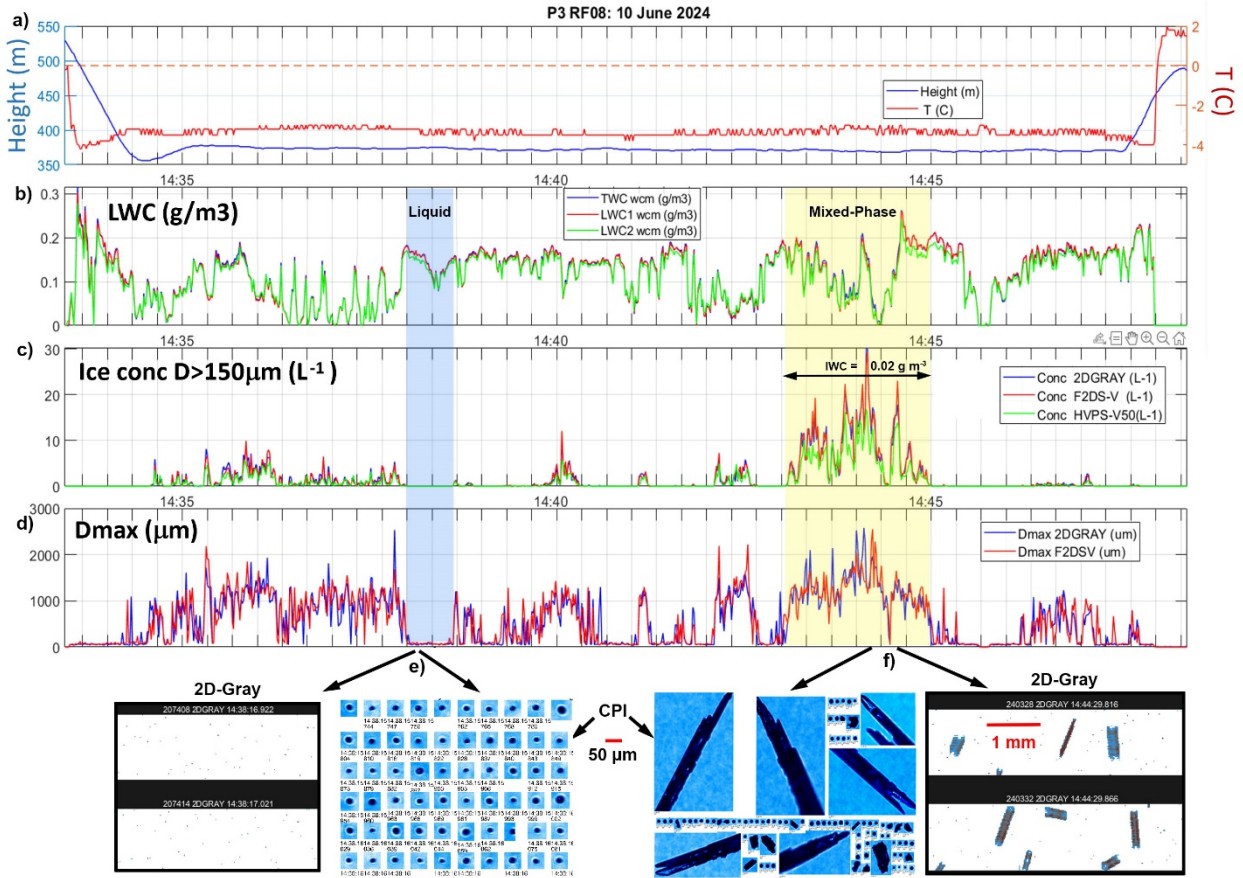

**Figure 6**. Time series measurements from P-3 RF08 flight on 10 June 2024 in a single-layer SBC showing a) Temperature and height, b) LWC from three WCM-2000 sensors, c) Concentration and d) maximum particle dimension of ice (> 150 µm) from 2D-Gray, 2D-S and HVPS4-50V probes, and average IWC using Baker and Lawson (2006) shown in the mixed-phase region. Examples of images from 2D-Gray and CPI probes in e) region with all-liquid cloud drops < 50-µm shaded in blue, and f) mixed-phase region in yellow.

Figure 8 shows another example of the cellular structure of ice in a SBC observed by the P-3 on 7 June 2024 (RF07). In this case the P-3 is flying in ice precipitation at 110 m (-1.8° C) below cloud base, which is at 260 m (-3.5° C) and cloud top at 470 m (-5° C). There is a strong correlation in Fig. 8 between in situ ice concentration, IWC, extinction coefficient and remote LSR (Lidar Scattering Ratio) nadir measurements at a spatial scale of 1 to 6 km from the Multi-function Airborne Raman Lidar (MARLi) (Wang et al., 2023). Note that the microphysical inhomogeneity in Fig. 8 is quasi-periodic and has a longer spatial scale than the intermittent inhomogeneity seen in Fig. 7. The observation of "cellular" structure (i.e., pockets of mixed-phase) near cloud tops, progressing through the cloud depth and into ice precipitation below, is suggestive that the environmental conditions favorable for primary ice initiation and/or secondary ice production are associated with those found inside these cells. While it is tempting to determine whether the cellular structure is associated with vertical air velocity, we point out that variations in vertical air velocity in ARCSIX clouds were typically ~ ±1 m s⁻¹, which is on the order of the measurement uncertainty. It is also worth

 noting that the cellular structure of ice formation in mixed-phase clouds was observed in previous studies (Luke et al., 2021; Shupe et al., 2008).

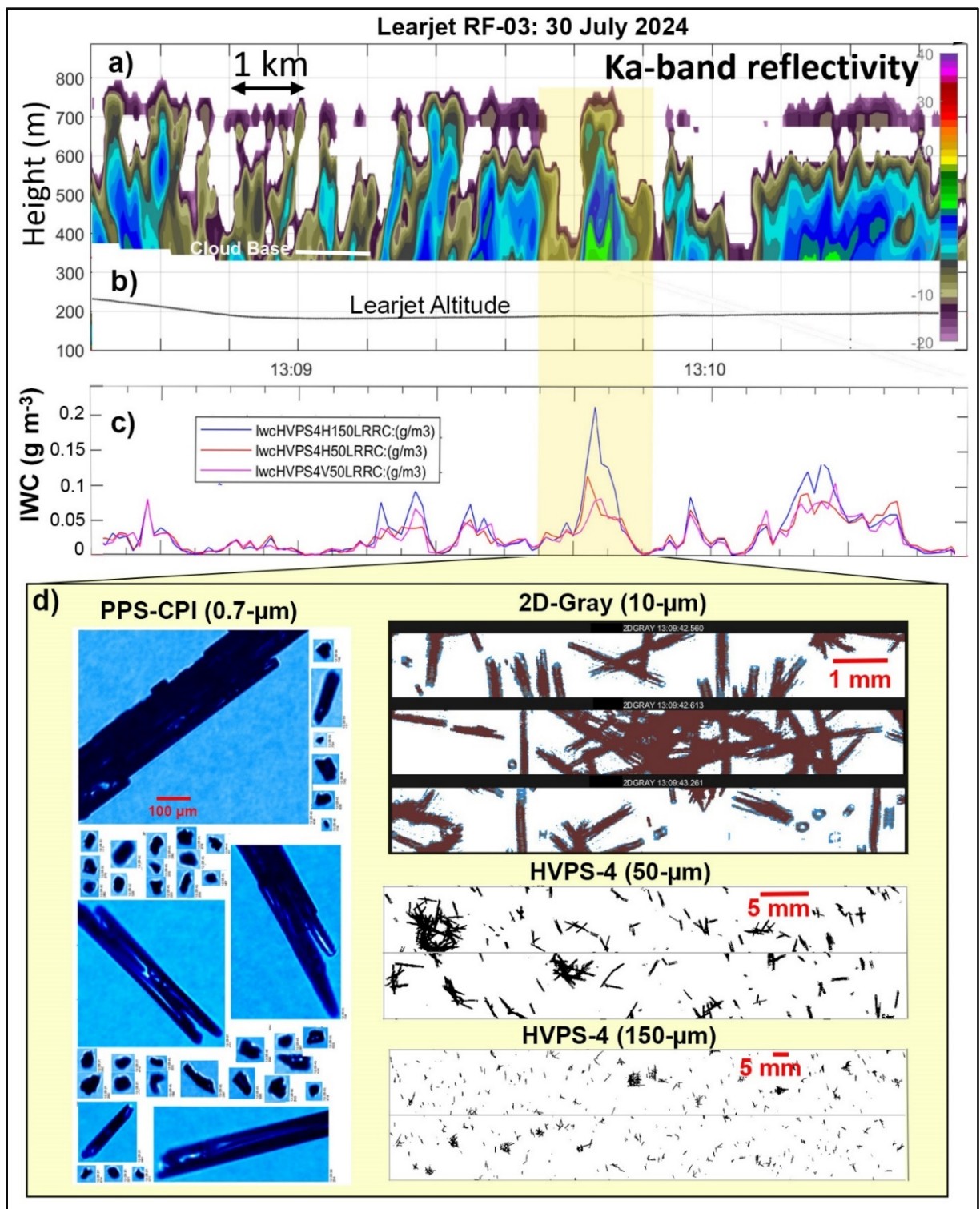

**Figure 7.** Time series from Learjet RF03 flight on 30 July 2024 showing: a) cellular structure in Ka-band radar reflectivity, b) altitude, c) IWC from HVPS4 50 and 150 µm channels, and d) examples of particle images below cloud base from 0.7-µm pixel-size CPI, 2D-Gray and HVPS4 probes shown in yellow shaded area (130935 – 130950).

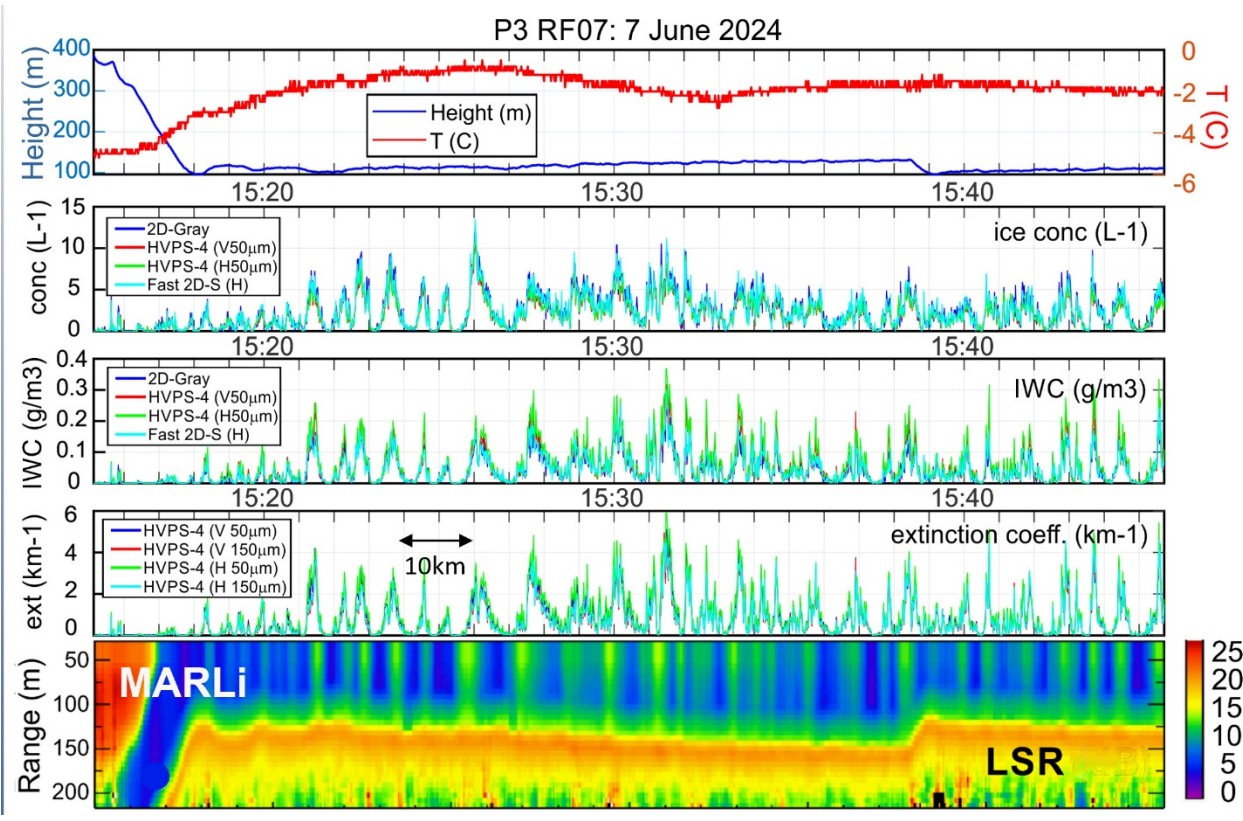

**Figure 8**. Time series from P-3 RF07 flight below cloud base on 7 June 2024: a) Temperature and height, b) ice concentration, c) IWC for dimension > 150 μm, and d) extinction coefficient, showing phase correlation with e) MARLi Lidar Scattering Ratio (LSR). (MARLi data courtesy of Wang 2025 – Personal Communication).

## 4 Ice Development

### 4.1 Anomalous Ice Development in ARCSIX Clouds at "Warm" Temperatures

The example in Fig. 6 of initiation and development of the ice process in a single-layer mixed-phase cloud (P-3 RF08) with top temperature warmer than -4 °C is curious, but is not unique in airborne investigations of Arctic clouds. Forward video from the P-3 during RF08 shows scattered, thin cirrus above and in the vicinity of lower stratiform clouds. However, the cloud probes did not detect any ice particles in the clear air above cloud top. Also, a temperature inversion extended for 800 m from 0 to 2 °C above cloud, so any ice falling from above is unlikely to have survived falling into lower clouds.

INP filter measurements collected during the ARCSIX field campaigns (Perkins 2025 – Personal Communication) are compared in Fig. 9a with filter measurements from the MOSAiC (Multidisciplinary drifting Observatory for the Study of Arctic Climate) project (Creamean et al., 2022). Both ARCSIX and MOSAiC filters were processed offline using the immersion freezing technique as discussed in Barry et al. (2021). Overall, the INP concentrations from ARCSIX are significantly lower than the MOSAiC data also shown in Fig. 9a, but both datasets have the same trend and drop off precipitously in the region from -10° to -12 °C. However, the MOSAiC data show that onset freezing temperature extended to -6 °C, whereas there was no nucleation events in the ARCSIX dataset warmer than -12° C. Creamean et al., (2022) report that nucleation at $T$ < -10 °C predominantly occurred in July –

August, which is the same timeframe as the summer ARCSIX deployment. They also found that warm-temperature INPs contained proteinaceous material and was most prevalent over open water.

It is important to note that MOSAiC filters were exposed for 24 to 72 hours on the drifting ship, which is much longer than exposure times on the P-3, which was from 20 to 60 minutes. Overall, the ARCSIX measurements in Fig. 9a are in reasonable agreement with historical measurements of INP in the Arctic and Antarctic shown in Fig. 9b (Kanji et al., 2017). Thus, the origin and development of ice at -4° C in the P-3 RF08 case (and other ARCSIX cases) cannot be explained by primary nucleation, or seeding from colder clouds above.

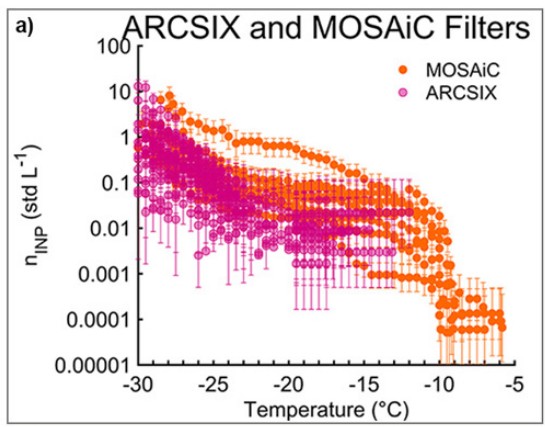 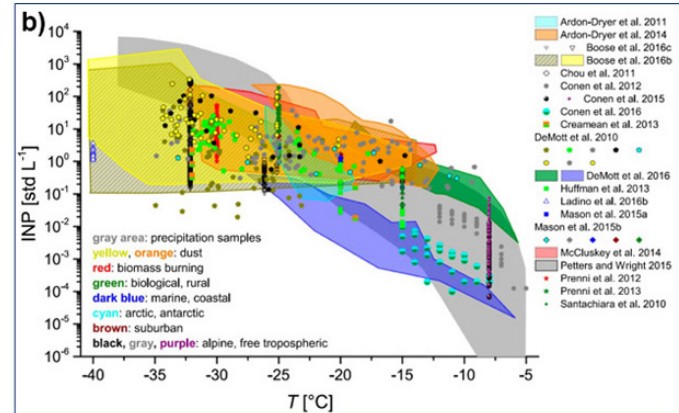

**Figure 9**. Plots of 31 ARCSIX filters collected below 10,000 feet between 5/28 - 6/13 and 7/25 - 8/15/2024 (Perkins 2025 – Personal Communication), compared with 8 filters collected near the surface on the R/V Polarstern as part of the MOSAiC expedition within the same time period, but in 2020 (Creamean et al., 2022), and b) historical INP measurements reported by Kanji et al., (2017).

In addition to the P-3 RF08 case shown in Fig. 6, there were additional ARCSIX single-layer SBCs that contained ice particles with cloud-top temperatures warmer than -4 °C. For example, the P-3 sampled a cloud layer with a (variable) cloud base at ~ -2 °C and cloud top at -3.9 °C on 6 June 2024 (RF06). The P-3 video shows a cirrus deck that is estimated to be at 6098 m with a temperature of -24 °C. On descent the P-3 cloud probes did not image any ice particles in the clear air. An all-liquid cloud deck 105 m thick with cloud-top temperature of -8.8 °C and a maximum LWC of 0.15 g m⁻³ was sampled from 132808 – 132834 UTC. The P-3 continued descent and recorded a ~ 500-m clear-air region with $0.5° \leq T \leq 1.5$ °C above a 194 m thick mixed-phase cloud deck with $-3.8° \leq T \leq -2.2$ °C sampled from 133958 – 135448 UTC. Any ice particles falling from above would have melted and evaporated in the 500-m layer with $0.5° \leq T \leq 1.5$ °C above the mixed-phase cloud deck.

Figure 10 shows time-series measurements and particle images from the penetration of the mixed-phase cloud deck described above during RF06: a) temperature and altitude, b) LWC from the FCDP and two channels of the WCM-2000, c) ice particle concentration $\geq 100$ µm from two 50-µm channels of the HVPS4, d) maximum ice particle dimension from the 2D-gray and 2D-S probes, and particle images $\geq 100$ µm in regions near 1343 and 1353 UTC from e) PPS CPI; f) 2D-Gray and g) HVPS4 50-µm H channel. As seen from the CPI images in Fig. 10e, cloud drops do not exceed about 40-µm in diameter in the region from 1340 – 1348 UTC where the HVPS4 detected copious columnar ice particles out to ~ 1 mm. Whereas, the region from 1348 – 1355 UTC contained drizzle drops out to about 100 µm and very rare detectable ice particles. The time-series data from the single-layer SBC in Fig. 10 show that a

region from 1340 – 1348 UCT contained small ($< \sim 40$ μm) cloud drops with 1-mm ice in concentrations up to nearly 10 $L^{-1}$, which was in juxtaposition with another region containing drizzle with virtually no ice. The scale of the inhomogeneity in this case was about 30 km. Inhomogeneity in single-layer ARCSIX SBCs was commonly observed, where scales ranged from 100's of meters to 100's of km. Inhomogeneity was observed in clouds that were nearly all mixed-phase with pockets of all-liquid regions, and conversely, in nearly all-liquid clouds that were interspersed with mixed-phase regions. The data in Fig. 10 show a maximum LWC of about 0.45 g $m^{-3}$, and a maximum ice particle concentration of nearly 10 $L^{-1}$. The ice concentration at $T \geq -3.8°$ in Fig. 10 exceeds INP measurements by nearly five orders of magnitude at $-6$ °C (which is the warmest temperature where INP measurements are available from Fig. 9). Thus, origin of the high ice particle concentration in the layer with $-3.8° \leq T \leq -2.2$ °C cannot be explained from our current theory of primary nucleation.

We do not have an explanation for how ice developed in the RF06 mixed-phase cloud with $-3.8° \leq T \leq -2.2$ °C shown in Fig. 10, nor in the example shown previously in Fig. 6. INP measurements do not support primary nucleation in this temperature range. Also, P-3 measurements below the level of the lower cloud decks did not reveal any ice particles lofted from the surface, which was mostly ice-covered with some open leads; however, due to sampling limitations this possibility cannot be dismissed. The anomalous occurrence of ice in Arctic SBCs at surprisingly warm temperatures ($\geq -4$ °C in Figs. 6 and 10) is not unique in the ARCSIX dataset. Ice was observed at temperatures $\geq -6$ °C on five other P-3 missions and two Learjet missions (see Table S1 and Table S2). As a counterpoint, all-liquid clouds were observed on several occasions down to temperatures of -14 °C, and the number of all-liquid clouds occurred approximately equally between spring and summer deployments (Table S1 and Table S2).

An example of another unusual case of ice development occurred on 8 August 2024 (Learjet RF07) during the summer deployment. Figure 11 shows a) representative 2D-Gray probe images, b) – d) time-series measurements from cloud particle probes, e) up/down Ka-band radar reflectivity measurements collected by the Learjet during a descent and level-off from 1500 m (-9.4 °C) to 190 m (-2.7 °C), and f) particle mass size-distribution from cloud particle probes. There was a brief region during the descent from 132420 - 132428 UTC (1,310 - 1,270 m) where the cloud probes did not detect any water drops or ice particles, which could suggest this was a multiple-layer cloud, or a region of clear air within the cloud.

The in-situ data in Fig. 11 indicate a 'mostly' all-liquid region with intermittent clear air from cloud top (1500 m, -9.4 °C) down to 1270 m (-8.1 °C). Drizzle drops 80 to 100-μm in diameter were observed in this layer increasing in diameter to about 300 μm near a ragged cloud base at about 190 m (-2.8 °C). Since the Learjet Ka-band radar is sensitive to the concentration of precipitation size particles (i.e., $\geq \sim 50$ μm), the radar cloud top suggests an estimate of the maximum altitude of drizzle that varies between 1200 m and 1600 m. Figure 11a shows examples of particle images as a function of time and temperature from the 2D-Gray and HVPS-4 50-μm channel scaled to 10-μm resolution.

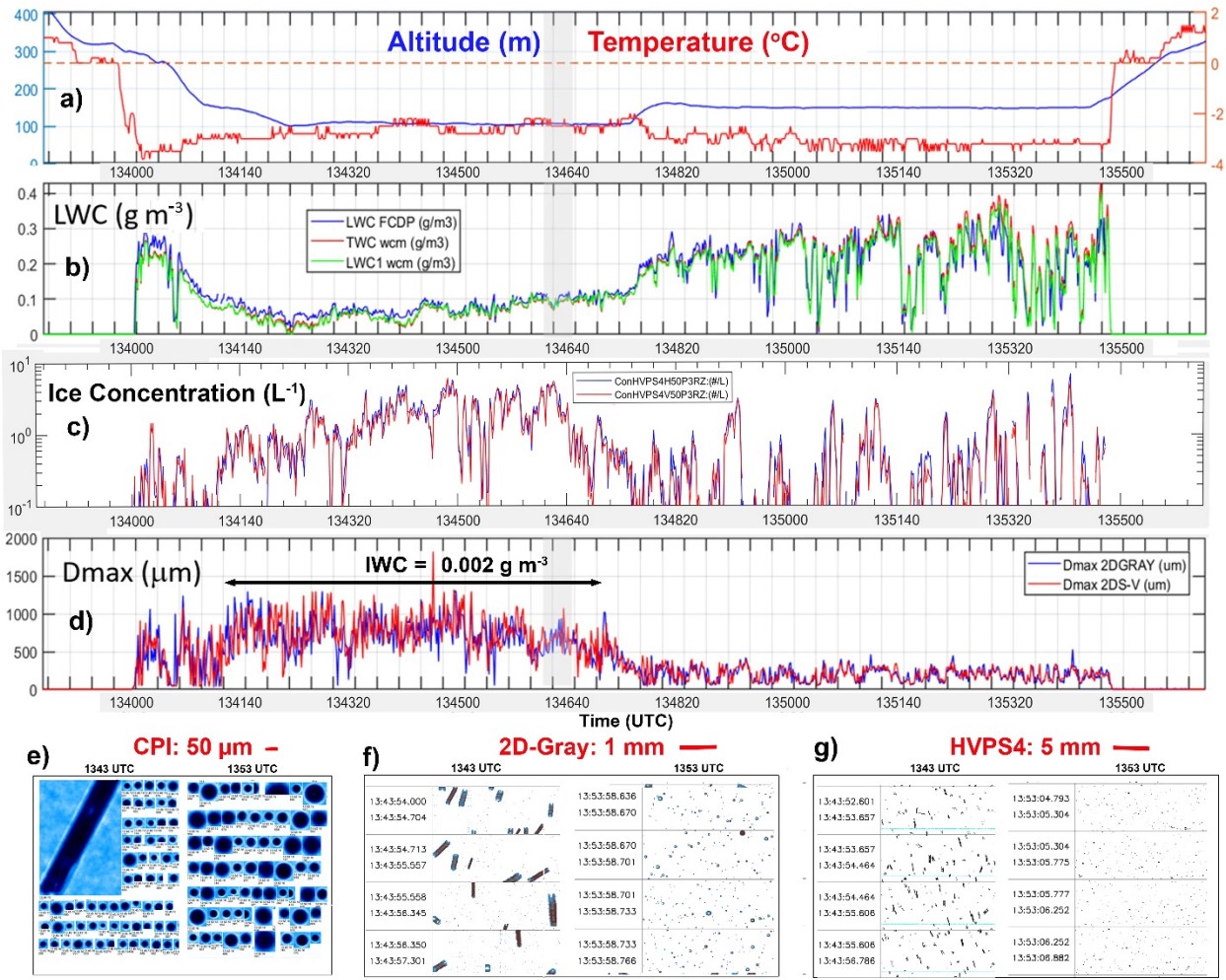

**Figure 10.** Time-series measurements from the P-3 6 June 2024 case (RF06): a) temperature and altitude, b) LWC from the FCDP and two channels of the WCM-2000, c) ice particle concentration ≥ 100 µm from two 50-µm channels of the HVPS4, d) maximum ice particle dimension from the 2D-gray and 2D-S probes and average IWC using Baker and Lawson (2006). Particle images in regions near 1343 and 1353 UTC from e) PPS CPI, f) 2D-Gray and g) HVPS4 50-µm H channel.

The first detectable ice particle, the only ice particle in the 'mostly' all-liquid layer, appeared to be a 200-µm frozen drop (graupel embryo) detected at 132410 UTC (1,365 m, -8.7 °C) on the H-Channel of the 50-µm HVPS-4 probe. Three more frozen drops 200- to 300-µm in diameter were detected in the descent down to 1,190 m (-7.7 °C), and a 550-µm graupel particle was observed on the HVPS-4 at 132447 UTC (1,126 m, -7.7 °C). A 650-µm graupel particle was observed on the 2D-Gray probe at 132502 UTC (1,015 m, -6.8 °C). The concentration of graupel particles was very small, on the order of < 1 m⁻³ in the layer between 1,015 and 1,365 m.

At 132520 UTC the first columnar ice particle was observed on the 2D-Gray at 872 m (-6.1 °C). The rapid increase in particle concentration from the 2D-S and 2D-Gray (Fig. 11c), and maximum particle size (Fig. 11d) during the descent from 132500 to 132550 UTC is primarily the result of increasing size and number of columnar ice particles. Drizzle drops also contributed to increasing particle concentration in this layer, but their maximum size did not exceed 300-µm, while the columns grew to millimeters in length. The particle images show that nucleation at – 6.1 °C was

followed mostly by diffusional growth during sedimentation of the ice. The columns were very rarely rimed with cloud drops, but when riming did occur, it was more often the result of rare drizzle drops frozen on the columns, which resulted in a lollipop appearance in the 2D-Gray images. Since riming on the columns was almost nonexistent, most of the increase in columnar size was via diffusional growth.

The presence of graupel particles was likely due to freezing of a (~ 100 μm) drizzle drop at ~ -8.7 °C. The relatively larger drop volume of drizzle increases the likelihood of immersion freezing compared with the much smaller cloud droplets (Pruppacher and Klett 2010). The graupel particles did not grow to sizes larger than about 1 mm, and were not observed below 500 m, whereas the columns grew to 2 mm and continued to cloud base and precipitated. No graupel was observed in the precipitation below cloud base. The very sparse concentration of graupel particles and absence of graupel in the precipitation below cloud is curious, but can possibly be explained from the drop-size distribution of mass (Fig. 11f). The fall velocity of small (< 500 μm) graupel is of the same order as 300-μm drizzle drops, inhibiting the potential for graupel – drizzle collisions. Also, as shown in Fig. 11f, the mass of drizzle drops (0.16 g m$^{-3}$) is twice the mass of cloud drops (0.08 g m$^{-3}$), limiting collisions and accretional growth of graupel expected from the differential in the two particle fall velocities.

The curious aspect of ice formation in this cloud is that, copious columnar ice crystals formed at – 6.1 °C in the middle of the cloud layer, with essentially all-liquid cloud above from – 6.1° to -9.4°C. This observation seemingly conflicts with the current theory of ice nucleation, which suggests a decrease of primary ice nucleation with increasing temperature, but is consistent with the overall trend seen in ARCSIX SBCs (i.e., Fig. 5). As described in the Introduction, it is also worth noting that liquid cloud above (and below) mixed-phase cloud in Arctic SBCs is not a unique observation and has been reported previously (e.g., Lawson and Zuidema, 2009; Lawson et al., 2011; Mioche et al., 2017; Järvinen et al., 2023).

An obvious challenge in our understanding of ice development from aircraft observations in ARCSIX clouds stems from the limited sample volumes of the particle probes. As was previously shown in Figs. 6, 7, 8 and 10, ARCSIX SBCs can be extremely inhomogeneous, and a slant descent through a mixed-phase region provides a fractional picture of the entire cloud. Previous and subsequent aircraft penetrations within this same SBC revealed all-liquid conditions. Thus, the data in Fig. 11 may represent continuously changing cloud conditions that are influenced by surface properties (e.g., nucleation by INPs or ice lofted into cloud), dynamics, or other transient environmental factors.

### 4.2  Secondary Ice Production

### 4.2.1  Learjet RF07 Case

ARCSIX SBCs contained a large range of ice particle concentrations and microphysical conditions. Some clouds contained ice concentrations orders of magnitude higher than predicted from primary nucleation, but did not exhibit recognized microphysical conditions that support SIP. On the other hand, some clouds technically satisfied conditions for SIP, but either there was no evidence of SIP, or the SIP conditions appeared to be too marginal to generate an active SIP process, or the SIP process was in its initial stage and had yet to produce higher than expected ice particle concentrations. Just as ARCSIX clouds did not always follow the expected prediction of ice as a function

of temperature from primary nucleation, ARCSIX clouds often did not behave in accordance with our current understanding of SIP processes.

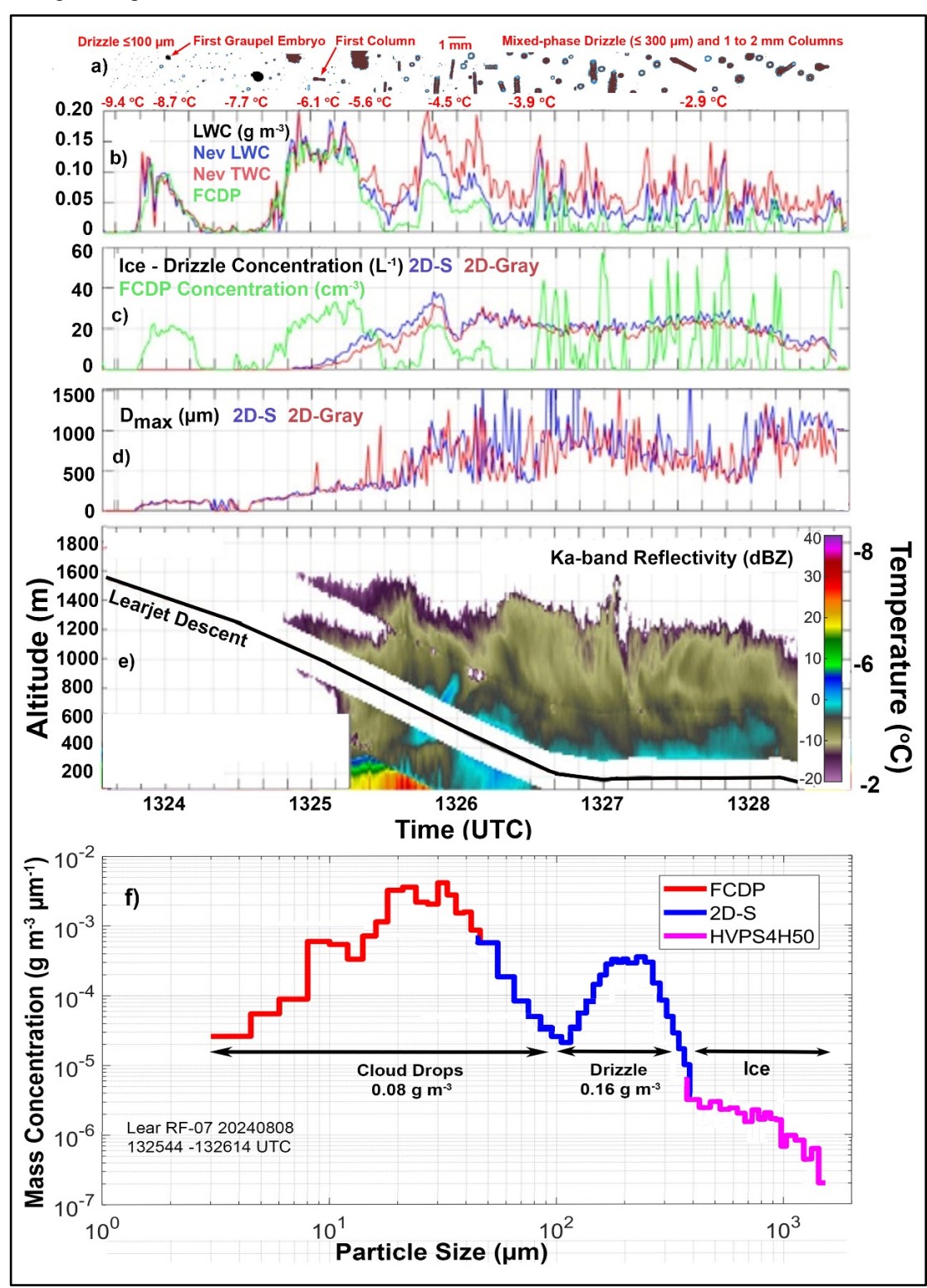

**Figure 11.** Time-series measurements from Learjet RF07 mission on 8 August 2024: a) representative 2D-Gray probe images, b) – d) time-series measurements from cloud particle probes, e) up/down Ka-band radar reflectivity measurements, and f) mass particle size distribution separated into water, drizzle and ice.

The microphysical conditions in the 8 August 2024 RF07 cloud that the Learjet investigated (Fig. 11), may, or may not have been conducive for SIP. The concentration of columnar ice particles in the middle region of cloud exceeded 30 L$^{-1}$, which is five orders of magnitude greater than predicted from primary nucleation at -2 ° to -6°C. (Fig. 9). The two likely SIP candidates are the Hallett-Mossop (HM) process (Hallett and Mossop, 1974), and the fragmentation of frozen drops (FFD) (Koenig, 1963; Lawson et al., 2015; Phillips et al., 2018; Korolev et al., 2024).

The generally accepted conditions for the "rime splintering" HM SIP are -8° ≤ T ≤ -3 °C, drops ≥ 23 μm in concentrations ≥ 1 cm$^{-3}$, the presence of drops <13 μm, and the presence of large, rimed ice particles (called "rimers"). The original definition of rimers was ice particles ≥ 300 μm, and the rate of ice splinter production was a maximum at a terminal velocity of 2.5 m s$^{-1}$ (Hallett and Mossop, 1974; Mossop and Hallett, 1974). However, in later laboratory investigations the range of terminal velocities was expanded by Saunders and Hoseini, (2001), who found a maximum production rate at 6 m s$^{-1}$. ARCSIX SBCs with $T \geq$ - 6 °C were composed almost exclusively of unrimed needles, sheaths, and columns (Figs 6, 7, 10). 1-mm needles and sheaths have a maximum terminal velocity of about 0.3 m s$^{-1}$ and columns have a maximum terminal velocity of about 0.9 m s$^{-1}$ (Heymsfield, 1972), which is well below the maximum production rate found in the laboratory studies. Also, as shown in CPI and 2D-Gray probe images in Figs. 6, 7 and 10, there is no evidence of riming on the columns, needles and sheaths. As discussed in the review by Korolev and Leisner, (2020), more recent laboratory experiments show that a buildup of a rimed surface on the rimer promotes the production of splinters, and this tends to occur during the formation of graupel. In the absence of graupel and riming on large columnar ice particles, we conclude that SIP from the HM rime-splintering process was likely nonexistent in the cases cited above.

The presence of graupel was virtually nonexistent in the RF07 case and there was no evidence of riming on the large columnar ice particles. At the Learjet true airspeed of 115 m s$^{-1}$, the two 50-μm channels of the HVPS-4 probe have a combined sample volume of 36 m$^{-3}$ for the descent and level-out from 1325 to 1328 UTC. The probe recorded 20 graupel particles during this time period, which equates to a graupel concentration of 0.55 m$^{-3}$. On the other hand, unrimed columnar ice particles were observed in concentrations on the order of 10,000 m$^{-3}$. A minimum concentration of graupel to support HM has not been determined, but in this case the graupel concentration appears to be too low to produce significant SIP via the HM process. It would appear in this case that the very low concentration of graupel along with the absence of riming on the columnar ice prevented the HM SIP from operating.

Conditions for the FFD SIP are not as well defined as for HM SIP. FFD SIP has been observed in-situ in conjunction with strong coalescence and millimeter-diameter supercooled large drops (SLDs) in cumulus clouds. These observations extend over several field campaigns and geographic locations (Lawson et al., 2015, 2017, 2022, 2023). In-situ observations of the fracturing of supercooled drizzle drops have not been investigated as extensively. Korolev et al., (2020) investigated a tropical mesoscale convective system with supercooled drizzle (diameter less than 300 μm). They found evidence of FFD SIP above the melting layer and surmised that the fractured frozen drizzle drops were ice particles that had previously melted and were brought upward in convective updrafts.

Lawson et al., (2015) reported that the production of secondary ice increased exponentially as the diameter of supercooled drops increased from hundreds (drizzle) to thousands of microns (raindrops). Keinert et al., (2020) performed laboratory experiments of the freezing characteristics of 300-μm drizzle drops with and without an aqueous

solution (2.9 mg L$^{-1}$) of sea salt analog (SSA) in a moist airflow. Their results showed that the FFD frequency of occurrence of drops with SSA was zero at temperatures warmer than -8 °C. For pure water drops there was a low, but measurable frequency of FFD in the temperature range of -10° ≤ T ≤ -2°C, which is the range where FFD was observed by Korolev et al., (2024). The drizzle drops in the mesoscale convective system investigated by Korolev et al., (2024) between 4700 and 7200 m were likely close to pure water. On the other hand, the RF07 cloud was sampled between 200 and 900 m in the temperature range of -6.2° ≤ T ≤ -2.5 °C over the Lincoln Sea with melted ice, so the drizzle drops were more likely to contain SSA, which may explain the lack of FFD SIP. PPS CPI images and images from both 2D-Gray probes failed to show significant existence of fractured supercooled drops. Thus, it is unlikely that the FFD SIP was active in this case.

### 4.4.2 P-3 RF16 Case

Mixed-phase regions of ARCSIX SBCs were often within the HM SIP temperature regime (-8° ≤ T ≤ -3 °C). While some of these cloud regions satisfied our HM SIP criteria, i.e., they contained graupel and exhibited what appeared to be high concentrations of columnar ice particles consistent with HM SIP, others that technically satisfied the HM SIP criteria contained very low concentrations of columns and much lower total concentrations of ice particles. To exemplify this, we examine a single-layer cloud sampled by the P-3 flight on 7 August 2024 (RF16) with a ~10 km all-liquid region bookended by two pockets of mixed-phase cloud. The cloud base temperature was about -4. 0 °C and the cloud top temperature was about -6.7 °C, although both cloud base and top temperatures were variable.

Figure 12 shows timeseries measurements of temperature, altitude, microphysical characteristics and representative images of cloud particles from RF16. The two cloud regions delineated in Fig. 12d are: *Region 1*) from 142428-142648 UTC at T = -6.2 °C, cloud drop concentration = ~ 80 cm$^{-3}$, LWC = ~ 0.3 g m$^{-3}$, 0.53 L$^{-1}$ of up to 4 mm graupel, 10 L$^{-1}$ columns up to 0.6 mm in size, and a concentration of ice particles (≥ 150 μm) up to 20 L$^{-1}$; and *Region 2*) from 143000-143130 UTC with very similar environmental conditions: T = -6.4 °C, cloud drop concentration = ~ 80 cm$^{-3}$, LWC = ~ 0.3 g m$^{-3}$, but with smaller (1 to 1.5 mm) graupel that is an order of magnitude less in concentration (0.013 L$^{-1}$) , two orders of magnitude fewer columns (0.1 L$^{-1}$) and 5 L$^{-1}$ ice particles (≥ 150 μm). In the 10-km between these two regions the single-layer cloud was all-liquid region with T = -6.3 °C, a cloud-drop concentration = ~ 80 cm$^{-3}$, LWC = ~ 0.3 g m$^{-3}$, and drizzle drops 60 – 250 μm in a concentration of 10$^{-2}$ L$^{-1}$.

Both of these cloud regions meet the *basic* requirements for HM SIP, i.e., both regions are within the HM temperature range, there is a presence of graupel, drops in excess of 1 cm$^{-3}$ with diameters ≥ 26 μm and ≤ 13 μm. However, *Region 2* had one-sixth the number of ice particles ≥ 150 μm and two orders of magnitude fewer columns than *Region 1*, with no evidence of riming, both of which are signatures of HM SIP. We consider two possibilities for this inconsistency. One possibility is that the production rate of HM SIP in *Region 2*, a microphysical process that is not fully understood, is much less than typically reported in the literature (e.g., 350 particles per milligram of accreted rime – Hallett and Mossop, 1974). Seidel et al., (2024) carefully repeated the HM laboratory experiments using high-speed videography and thermal imaging. They found no evidence of an efficient rime-splintering mechanism, with some experiments resulting in no SIP at all. However, this does not adequately explain why *Region 1* contained

concentrations and characteristics of ice particles consistent with HM SIP as reported in Hallett and Mossop, (1974),

and *Region 2* did not.

A second possibility is that lack of columnar ice and lower total ice particle concentration is due to the relatively low concentration and size of the graupel particles in *Region 2* compared with *Region 1*. The HM mechanism implies that the concentration of rime splinters (i.e., secondary ice) is positively correlated with the number and size of graupel particles. However, the actual production rate of secondary ice as a function of graupel size and

concentration is unknown, and could be zero under some conditions that apparently qualify for HM SIP (e.g., Seidel et al., 2024).

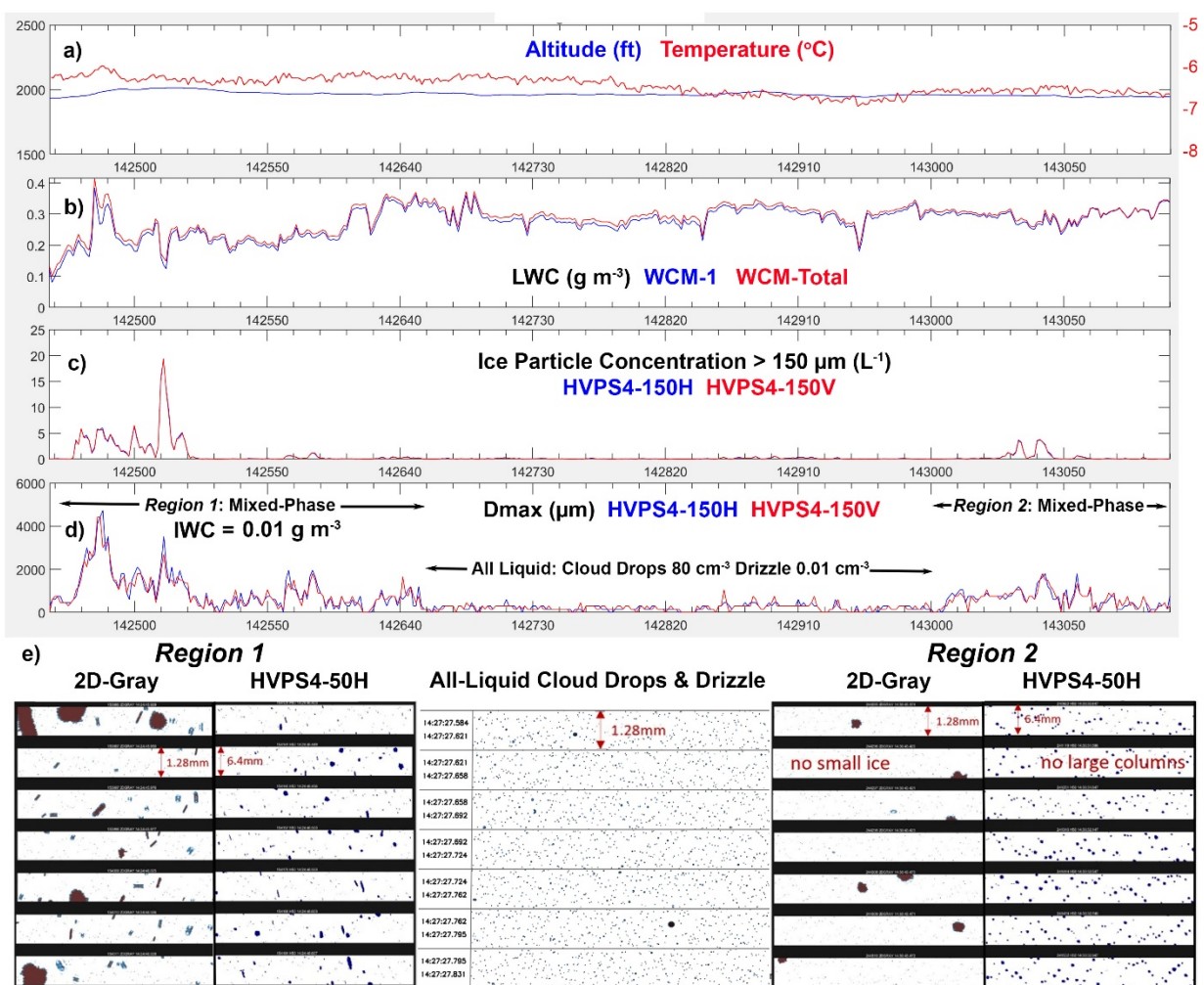

**Figure 12**. Time-series measurements from P-3 RF16 flight on 7 August 2024 of a) Altitude and temperature, b) LWC from two WCM-2000 sensors, c) ice particle concentration ≥ 150 μm from two HVPS4 150-μm channels, d) maximum

particle dimension from two HVPS4 150-μm channels and average IWC using Baker and Lawson (2006) shown in mixed-phase region and e) examples of particle images from 2D-Gray and HVPS4 50- μm channels in *Region 1*, *Region 2*, and the all-liquid region with cloud drops and drizzle in between.

Figure 13 shows a comparison of the particle size distributions (PSDs) from *Regions 1* and *2*. The obvious difference in the PSDs is the much higher concentration of particles > 0.5 mm, which are determined to be graupel

from the particle images. Conditions for HM SIP found in the literature require the presence of graupel, but there is

no quantitative evidence specifying the minimum required concentration and size of the graupel particles. Thus, HM SIP could be occurring in *Region 2* at a much lower rate than *Region 1*. Another possibility is that the HM process in *Region 2* may be in its embryonic stage compared to *Region 1*, and at a later point in time *Region 2* may display similar ice particle characteristics as *Region 1*. A third possibility is that SIP was occurring in *Region 1*, but is not one of the six mechanisms listed in Korolev and Leisner, (2020), and is not currently understood.


We also point out here that "snapshot" samples collected by aircraft measurements cannot provide a Lagrangian perspective of the development of cloud microphysics. Even with return passes through the same cloud region (which was not an objective of this campaign), data collection is time and spatially aliased, providing only snapshots of a continuous, time-dependent process.

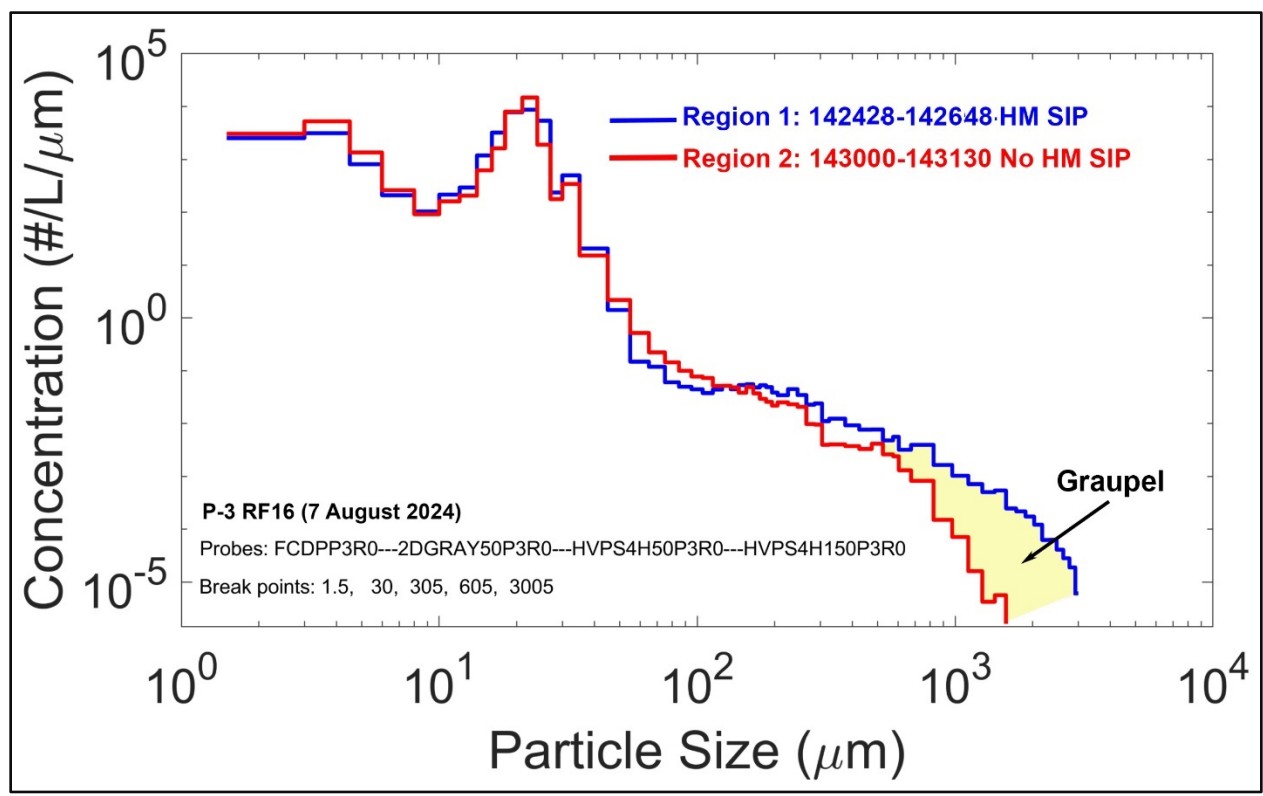


**Figure 13**. Composite (FCDP, 2D-Gray, HVPS4-50 and HVJPS4-150) size distributions for *Region 1* and *Region 2* shown in Fig. 12.

## 5   Summary and Discussion

The P-3 and Learjet flew a total of 30 missions during the spring (30 May – 13 June 2024) and summer (24 July – 16 August 2024) ARCSIX deployments. A total length of 12,417 km clouds were sampled by the two aircraft using in situ and/or remote sensors. Out of that total 6,966 km of the clouds were investigated in situ with a suite of state-of-the-art microphysical cloud probes installed on both aircraft. The focus of this paper is to investigate the microphysical properties of stratiform boundary-layer clouds (SBCs). Mixed-phase SBCs were sampled during 60.5% of time in cloud, and all-liquid SBCs were encountered 39.5% of the time. The clouds displayed considerable inhomogeneity, with all-liquid regions of small (< 30 μm diameter) cloud drops, all-liquid regions with small cloud drops and drizzle drops, regions with small cloud drops and ice particles, and regions with small cloud drops, drizzle



and ice particles. While inhomogeneity in SBC's was frequently observed, there were infrequent occasions where cellular structure was observed on scales ranging from 100's of m to 10's of km (e.g., Figs. 7 and 8), which could not be explained by convection or wave dynamics. That said, while SBC cloud tops were almost always smooth and vertical motions were small (typically $< \pm 1$ m s$^{-1}$), a thorough quantitative investigation of vertical motions in ARCSIX has yet to be performed.

About 90% of the SBCs had cloud-top temperatures $\geq$ - 9 °C, and on several missions, ice was imaged in clouds with top temperatures $\geq$ - 6° C. In two case studies presented in this paper (Figs. 6 and 10), examples of mixed-phase clouds are analyzed with cloud-top temperatures $\geq$ - 4° C that indicate no observable possibility of ice-particle seeding from colder clouds aloft. Measurements during ARCSIX (and in all other campaigns in the Arctic) do not support the existence of significant concentrations of INPs that are active at temperatures $>$ - 6°, which begs the question: How did ice form at these relatively warm temperatures?

It is well-known that Arctic SBCs can persist for days at time (Intrieri et al., 2002; Morrison et al., 2012, Shupe et al. 2008). The classical theory of nucleation addresses thermodynamic and kinetic factors, including the time-dependence of INP activation (Pruppacher and Klett, 2010). While observations of time-dependent freezing abound, rigorous in-situ projects quantifying the time variable are absent. That said, the extremely long lifetime of Arctic SBCs may be an indicator that given adequate duration, ice can form in Arctic clouds at temperatures warmer than predicted from current INP measurements. The MOSAiC measurements from a drifting ship, where filters were exposed for ~24 hours at a time, are the only measurements showing INPs active at - 6° C. Presumably, this is due to the long capture time of a very rare INP that is active at this temperature. However, even if very rare INPs active at $T >$ - 6° C are responsible for initiating ice in some long-lived ARCSIX clouds, there still remains the unanswered questions of how high concentrations of ice are generated, and how large ice particles persist in shallow mixed-phase SBCs.

One possibility that could produce primary nucleation in SBCs with $T >$ - 6° C depends on the existence of very rare (and currently unmeasured) INPs that are active at $T >$ -6 °C. If such INPs exist and are ingested into SBCs to produce very sparse ice particles, the ice particles will eventually collide with and freeze other supercooled drops. If the collision results in a (albeit unlikely) hypothetical process where the supercooled drop freezes, but does not stick to the ice particle, then the concentration of ice particles will increase. This hypothetical process is similar to contact nucleation (Cooper, 1974), where an INP interacts with the surface of a supercooled drop, but does not become ingested into the frozen drop. In this way, ice particles can increase in concentration and grow in size via the Wegener-Bergeron-Findeisen process. Since mixed-phase SBCs are observed to exist for hours and even days, this process could lead to formation of mixed-phase clouds with large concentrations of ice particles. However, even this hypothetical process does not explain how the ice in SBCs can grow to millimeter sizes and persist for days.

Yang et al., (2015) performed a large-eddy simulation (LES) of the growth of an ice particle in a 200-m thick Arctic SBC. They found that a 10-µm ice particle that was initiated at cloud top and fell through a quiescent, mixed-phase cloud, would grow to a maximum dimension of 200 µm at cloud base. In contrast, ARCSIX observations show columnar ice crystals 1 to 5-mm in length precipitating from the base of clouds that are only 250 m thick (Fig. 7). The terminal fall velocity of a 3-mm column is about 1.5 m s$^{-1}$ (Mitchell, 1996; Heymsfield and Westbrook, 2010), so these large ice particles should fall from cloud within a few minutes unless recycling is occurring.

Recycling was found to be significant in LES of a single-layer SBC observed during the Mixed-Phase Arctic Cloud Experiment (M-PACE) (Verlinde et al., 2007; Fan et al., 2009; Fridland and Ackerman 2018). Solomon et al., (2015) explain that radiative cooling near cloud top generates turbulence that maintains the liquid layer and forms an approximately well-mixed layer that extends as far as 500m below cloud base. The cloud-driven mixed layers are frequently decoupled from the surface layer, limiting the impact of fluxes of heat, moisture, and aerosols on the cloud layer from below (Solomon et al., 2011; Shupe et al., 2013). LES simulations performed by Solomon et al., (2015) suggest that sustained recycling of INPs through a drying subcloud layer and additional activation of new INPs due to diurnal cooling at cloud top are sufficient to maintain ice production over multiple days. Recycling within and below cloud provides a possibility were smaller ice particles can eventually grow to large sizes, and new INPs can be introduced to generate new ice particles.

Assuming that recycling is an active process in some Arctic SBCs, without another explanation, primary nucleation of clouds with $T > -6$ °C would first require rare INPs that are active at this warm temperature. However, another possibility is that the 'warm' cloud had a colder cloud top at some point in its history, which seems likely from diurnal cooling at cloud top given the long lifetime of SBCs. Assuming this hypothesis, ice particles precipitating from clouds with colder top temperatures (say $T < -10$ °C) could recirculate from below cloud base and be ingested into updrafts feeding clouds whose tops had subsided to a warmer temperature. The updraft velocities in and below Arctic SBCs are typically weak, and therefore difficult to measure with either aircraft or ground-based Doppler radar, but are typically be $< 0.5$ m s$^{-1}$ (Shupe et al., 2008). Thus, precipitating ice particles would have to sublimate in the dry mixed-layer below cloud base before being transported upwards.

It is possible that an ice particle could melt or sublimate except for a submicron "pit" of ice that remains in the original cloud condensation nucleus (CCN), forming a preactivated INP. Fournier d'Albe (1949) was the first to describe preactivation and perform expansion chamber experiments. Mossop (1956) repeated Fournier d'Albe's setup with 50 substances and found that only four exhibited preactivation; interestingly, one of the four was $CaCO_3$ (Iceland Spar). Marcolli (2017) provides an excellent review of preactivation nucleation from the early experiments through 2016. In ARCSIX SBC's a tiny residual aerosol particle with a submicron ice pit could then be entrained into a very low-velocity updraft and be ingested into a supercooled liquid cloud. While there is no way for current instrumentation to measure these "ice pit" particles in situ, the possibility that they do exist, and could seed a supercooled liquid cloud, is not zero.

As cogently pointed out by a reviewer, the current technology for measuring INPs as a function of temperature may not be adequate to explain how primary nucleation actually takes place in natural clouds. Collection of aerosols on filters followed by offline immersion freezing obviously introduces artificial factors that do not exist in natural clouds. Airborne INP processing using a Continuous Flow Diffusion Chamber (CFDC) attempts to simulate cloud conditions, but cannot accurately simulate repeated cycling of INPs or time dependence. It may be possible for large cloud chambers to simulate repeated cycling of INPs, but this has not been attempted at $T \geq -4$ °C. The above discussion is not intended to be a slam on current INP measurement methodologies. The purpose of this critique is to point out that in-situ cloud particle measurements and laboratory simulations cannot possibly encompass the complex physical and chemical interactions that impact primary nucleation in natural clouds.

Another possibility for explaining the origin of ice in SBCs with cloud-top $T \geq$ -4 °C is that ice particles were lofted from the ice surface into supercooled cloud above. Both the Learjet and P-3 flew below non-precipitating cloud at 100 to 200 m above sea level on multiple occasions. Images of ice particles were not observed on any of the cloud particle probes during these flights. However, the sample volumes of the cloud probes limit the probability of imaging small ice in very low (order 1 m$^{-3}$) concentrations and the time spent in this flight configuration was < 10% of the in-situ sampling time. That said, as seen in Fig. 6, the P-3 flew in a single-layer SBC with cloud-top $T$ = -4 °C over open ocean in the Baffin Sea and observed regions of mixed-phase cloud. This argues against the possibility of ice particles being lofted from sea ice as an explanation for the source of ice in warm ARCSIX clouds.

The large majority of ice particle habits in ARCSIX SBCs were columnar in shape, which is expected since a majority of the in-situ sampling was done between about -4° and -7 °C. However, both the CPI and 2D-Gray probes also imaged irregular-shaped particles in much smaller concentrations. The origin of irregular-shaped ice in an environment where the large majority of ice particles have grown via vapor diffusion, and there is little riming, is anomalous. There are two plausible explanations for the irregular ice particles. Large ice particles are known to shatter on the probe tips and inlets (Korolev et al., 2011; Lawson, 2011). However, another possibility is that these were ice particles that were shed from accumulated ice on the aircraft itself and/or on probe inlets/tips. This is possible since the dynamic temperature at stagnation points on the aircraft and probe inlets/tips is about 5 to 6° warmer than static temperature at the airspeeds of both the P-3 and Learjet in these conditions (Lawson and Cooper, 1990). Since a large portion of the ARCSIX dataset was collected in SBCs with $T$ > -6 °C, the leading surfaces of the aircraft and cloud probes could warm to $T \geq 0$ °C, enabling the shedding of accumulated rime ice. Other than these two possibilities, we do not have an explanation for the irregular-shaped ice particles that were imaged in SBCs where ice habits were characteristic of diffusional growth.

Ice concentrations were measured that were much higher than expected from primary nucleation, which suggests the possibility of SIP. The Hallet and Mossop, (1974) temperature criteria (-8° $\leq T \leq$ -3 °C) for SIP was encountered in over 90% of the in-situ observations of ARCSIX SBCs. This provided an excellent testbed for determining the observational conditions and frequency of HM SIP. While ARCSIX clouds often contained conditions that met all of the HM SIP criteria, others did not, but still contained high concentrations of ice at $T \geq$ - 4° C (e.g., Figs. 6 and 10). In contrast, some clouds technically met HM SIP criteria and did not contain exceptionally high ice concentrations indicative of HM SIP (e.g., *Region 2* in Fig. 11). The recent laboratory work of Seidel et al., (2024) repeated the HM experiments using high-speed video and thermal imaging. They found no evidence of an efficient HM secondary ice production. Their results and our measurements from ARCSIX strongly suggest that quantification of the mechanism(s) and rate of HM SIP are still a work in progress.

While estimates for the rate of HM SIP found in the literature vary (e.g. Mossop, 1976; Seidel et al., 2024), there is no quantitative evidence on how the size and concentration of graupel impacts the development and rate of HM SIP. In one case study (P-3 RF16), we examined the concentration of ice particles in two regions of mixed-phase cloud (*Region 1* and *Region 2*) separated by an all-liquid SBC that contained drizzle (Fig. 12). Both mixed-phase regions contained nearly identical microphysics (temperature, drop concentration, drizzle, graupel and small columnar ice) that met HM SIP requirements, except that *Region 1* contained much higher concentrations of larger graupel and

small columnar ice than *Region 2* (Fig. 13). We hypothesize that Region 2 may have been in the formative stage of HM SIP, where the concentration and size of graupel particles was insufficient to produce significant HM SIP.

Another SIP process that could explain the high ice concentrations in SBCs is the fragmentation of frozen drops (FFD). FFD SIP occurs when supercooled drops freeze and produce fragments, and/or project tiny particles through extruded tubes called spicules (Korolev and Leisner, 2020). Lawson et al., (2015, 2017, 2022, 2023) show several examples of FFD images from measurements in convective cloud at $-20 \leq T \leq -12$ °C. Their data also suggest that the probability of FFD increases exponentially with drop diameter, reaching a maximum with mm-diameter

supercooled drops. On the other hand, Korolev et al., (2020, 2024) examined a recirculation process in a mesoscale convective system and determined that 300-μm drizzle drops could produce FFD within a temperature range of $-10°$ $\leq T \leq -2$ °C.  Keinert et al., (2020) performed lab experiments that showed that FFD was a maximum near about -15 °C, and that the tendency for FFD also decreased with increasing concentration of sea salt in the temperature range from $-10° \leq T \leq -2$ °C.  The concentration of sea salt was not measured in the study from Korolev et al., (2024). The

proximity of SBCs to the sea surface increases the likelihood that CCN containing sea salt was ingested into ARCSIX SBCs, thereby inhibiting FFD SIP.  Also, drizzle was observed in low-level ARCSIX clouds at $T > -6$ °C except for one case (Learjet RF07 on 9 August 2024) when it was observed at -19°C.  Thus, the environmental and microphysical conditions conducive for supporting the FFD SIP process were suboptimal in ARCSIX clouds.

While the in-situ investigations of ARCSIX SBCs create more questions than they answer, the extensive

measurements with state-of-the-art technology have provided more detailed insights into the microphysics of these clouds than heretofore possible. Additional studies that include turbulence measurements in and below SBCs may shed light on possible nucleation mechanisms in SBCs with $-2° \leq T \leq -6$ °C. Further lab experiments and analysis of ARCSIX cases may help to quantify conditions controlling HM SIP. Finally, numerical models, particularly Lagrangian simulations combined with aircraft observations, will help provide insights into primary and secondary

nucleation processes.

**Data Availability**. All of the aircraft data are available at the NASA archive located at https://asdc.larc.nasa.gov/project/ARCSIX

**Supplements.** A zip file of spreadsheets containing supplemental data are attached.

**Author Contributions**. Preliminary analysis of data and figures are based on an invited presentation given by Korolev at the AMS Annual meeting in New Orleans in January, 2025. Additional analysis of data, figures and text have been contributed by Lawson.


**Competing Interests.** The authors declare that they have no conflict of interest.

**Acknowledgements.** The authors would like to acknowledge the excellent guidance of NASA RSP Program Manager Dr. Hal Maring and Chief ARCSIX scientist Dr. Sebastian Schmidt from the University of Colorado. We are indebted

to Ivan Heckman (ECCC) and Parker Morris, Ted Fisher and Qixu Mo (SPEC) for spending long hours writing code, processing data in the field and operating instrumentation on the SPEC Learjet. We thank all of the ARCSIX participants who braved the Arctic weather and are too many to mention here. The pilots of the NASA P-3, Brian Bernth, Greg Jenkins, John Baycura and Rodney Turbak, and SPEC Learjet pilots, T. R. Vreeland and Bill J. Harris are commended for flying challenging flight plans under adverse conditions. We would also like to acknowledge and

thank Dr. Zhien Wang (Stony Brook University) for providing MARli lidar data, and Dr. Russell Perkins, Dr. Paul DeMott, Dr. Sonia Kreidenweis, Dr. Kevin Barry, Dr. Ryan Patnaude, and Camille Mavis (Colorado State University) for collecting and providing INP data shown in this paper.

**Financial Support**. Support for Korolev was provided by Environment and Climate Change Canada (ECCC).
Funding was provided for Lawson under Grant No. 80NSSC22K1771 from the U.S. National Aeronautics and Space Administration (NASA) Radiation Sciences Program (RSP).

**References.** Baker, B. and Lawson, R. P.: Improvement in Determination of Ice Water Content from Two-Dimensional Particle Imagery. Part I: Image-to-Mass Relationships, 45, 1282–2111,
https://doi.org/10.1175/JAM2398.1, 2006.

Barry, K. R., Hill, T. C. J., Levin, E. J. T., Twohy, C. H., Moore, K. A., Weller  Z. D.,  Toohey, D. W., Reeves, M., Campos, T., Geiss R., Fischer, E. V., Kreidenweis, S. M., and DeMott, P. J.,: Observations of ice nucleating particles in the free troposphere from western US wildfires. J. Geophys. Res., 126, e2020JD033752. https://doi.org/10.1029/2020JD033752, 2021.

Bowman, K. P.: Large-scale isentropic mixing properties of the Antarctic polar vortex from analyzed winds, J. Geophys. Res., 98, 23013–23027, https://doi.org/10.1029/93JD02599, 1993.

Cooper, W. A.: A Possible Mechanism for Contact Nucleation, J. Atmos. Sci., 31, 1832–1837, https://doi.org/10.1175/1520-0469(1974)031<1832:APMFCN>2.0.CO;2, 1974.

Curry, J. A.: Interactions among Turbulence, Radiation and Microphysics in Arctic Stratus Clouds, J. Atmos. Sci., 43,
90–106, https://doi.org/10.1175/1520-0469(1986)043<0090:IATRAM>2.0.CO;2, 1986.

Creamean, J.M., Barry, K., Hill, T.C.J., Hume, C., DeMott, P.D., Shupe, M. D, Dahlke, S., Willmes, S., Schmale, J., Beck, I., Hoppe, C. J. M., Fong, A, Chamberlain, E, Bowman, J., Scharien, R., and Persson, O.: Annual cycle observations of aerosols capable of ice formation in central Arctic clouds. Nat Commun 13, 3537 https://doi.org/10.1038/s41467-022-31182-x, 2022

Dergach, A. L., Zabrodsky, M., and Morachevsky, V. G.: The results of a complex investigation of the type st-sc clouds and fogs in the Arctic., Bull. Acad. Sci. USSR, Geophys. Ser., 1, 66–70, 1960.

Fan, J., Ovtchinnikov, M., Comstock, J. M., McFarlane, S. A., and Khain, A.: Ice formation in Arctic mixed-phase clouds: Insights from a 3-D cloud-resolving model with size-resolved aerosol and cloud microphysics, J. Geophys. Res., 114, 2008JD010782, https://doi.org/10.1029/2008JD010782, 2009.

Fournier D'Albe, E. M.: Some experiments on the condensation of water vapour at temperatures below 0 ∘C, Q. J. Roy. Meteor. Soc., 75, 1–14, doi:10.1002/qj.49707532302, 1949.

Fridlind, A. M., and  Ackerman, A. S.: Simulations of Arctic mixed-phase boundary layer clouds: Advances in understanding and outstanding questions, C. Andronache, Ed., Elsevier, 153–183, https://doi.org/10.1016/B978-0-12-810549-8.00007-6, 2018.

Gayet, J., Treffeisen, R., Helbig, A., Bareiss, J., Matsuki, A., Herber, A., and Schwarzenboeck, A.: On the onset of the ice phase in boundary layer Arctic clouds, J. Geophys. Res., 114, 2008JD011348, https://doi.org/10.1029/2008JD011348, 2009.

Hallett, J. and Mossop, S. C.: Production of secondary ice particles during the riming process, Nature, 249, 26–28,https://doi.org/10.1038/249026a0, 1974.

Herman, G. F. and Curry, J. A.: Observational and Theoretical Studies of Solar Radiation in Arctic Stratus Clouds, J. Climate Appl. Meteor., 23, 5–24, https://doi.org/10.1175/1520-0450(1984)023<0005:OATSOS>2.0.CO;2, 1984.

Heymsfield, A. J.: Ice crystal terminal velocities. Journal of the Atmospheric Sciences, 29, 1348-1357, DOI: 10.1175/1520-0469(1972)029<1348:ICTV>2.0.CO;2, 1972.

Heymsfield, A. J. and Westbrook, C. D.: Advances in the Estimation of Ice Particle Fall Speeds Using Laboratory and 740 Field Measurements, Journal of the Atmospheric Sciences, 67, 2469–2482, https://doi.org/10.1175/2010JAS3379.1, 2010.

Hobbs, P. V. and Rangno, A. L.: Microstructures of low and middle-level clouds over the Beaufort Sea, Quart J Royal Meteoro Soc, 124, 2035–2071, https://doi.org/10.1002/qj.49712455012, 1998.

IPCC.: *Climate Change: The IPCC Scientific Assessment*. J.T. Houghton, G.J. Jenkins, and J.J. Ephraums 745 (eds.). Cambridge University Press, Cambridge, UK, and New York, NY, USA, 1990.

IPCC,: *Climate Change 1995: The Science of Climate Change*. Contribution of Working Group I. Houghton, J.T., et al. (eds.). Cambridge University Press, Cambridge, UK, and New York, NY, USA, 1996.

IPCC,: *Climate Change 2001: The Scientific Basis*. Contribution of Working Group I Houghton, J.T., et al. (eds.). Cambridge University Press, Cambridge, United Kingdom and New York, NY, USA, 881pp, 2001.

IPCC,: *Climate Change 2007*. Contribution of Working Group III. B. Metz, B, et al. (eds), Cambridge University Press, Cambridge, United Kingdom and New York, NY, USA, 2007.

IPCC,: *Climate Change 2013: The Physical Science Basis*. Contribution of Working Group I. [Stocker, T.F., et al. (eds.). Cambridge University Press, Cambridge, United Kingdom and New York, NY, USA. 2013.

IPCC,: *Climate Change 2023: Synthesis Report.* Contribution of Working Groups I, II and III to the Sixth Assessment 755 Report of the Intergovernmental Panel on Climate Change [Core Writing Team, H. Lee and J. Romero (eds.)]. IPCC, Geneva, Switzerland, pp. 35-115, doi: 10.59327/IPCC/AR6-9789291691647, 2023.

Järvinen, E., Nehlert, F., Xu, G., Waitz, F., Mioche, G., Dupuy, R., Jourdan, O., and Schnaiter, M.: Investigating the vertical extent and short-wave radiative effects of the ice phase in Arctic summertime low-level clouds, Atmospheric Chemistry and Physics, 23, 7611–7633, https://doi.org/10.5194/acp-23-7611-2023, 2023.

Kanji, Z. A., Ladino, L. A., Wex, H., Boose, Y., Burkert-Kohn, M., Cziczo, D. J., and Krämer, M.: Overview of Ice Nucleating Particles, Meteorological Monographs, 58, 1.1-1.33, https://doi.org/10.1175/AMSMONOGRAPHS-D-16-0006.1, 2017.

Keinert, A., Spannagel, D., Leisner, T., and Kiselev, A.: Secondary Ice Production upon Freezing of Freely Falling Drizzle Droplets, Journal of the Atmospheric Sciences, 77, 2959–2967, https://doi.org/10.1175/JAS-D-20-0081.1, 765 2020.

Klein, S.A., Mccoy, R.B., Morrison, H., Ackerman, A.S., Avramov, A., Boer, G.D., Chen, M., Cole, J., Del Genio, A. D., Falk, M, Foster, M.J., Fridlind, A., Golaz, J.-C. Hashino, T., Harrington, J. Y., Hoose, C., Khairoutdinov, M. F., Larson, V. E., Liu, X., Luo, Y., McFarquhar, G. M., Menon, S., Neggers, R. A. J., Park, S., Poellot, M. R., Schmidt, J. M., Sednev, I., Shipway, B. J., Shupe, M. D., Spangenberg, D. A., Sud, Y. C., Turner, D. D., Veron,

D. E., von Salzen K., Walker, G. K., Wang, Z., Wolf, A. B., Xie, S., Xu, K-M., Yang, F., Zhang, G.,: Intercomparison of model simulations of mixed-phase clouds observed during the ARM Mixed-Phase Arctic Cloud Experiment. I: single-layer cloud. Q. J. R. Meteorol. Soc. 135, 979–1002. https://doi.org/10.1002/qj.416, 2009.

Koenig, L. R.: The Glaciating Behavior of Small Cumulonimbus Clouds, J. Atmos. Sci., 20, 29–47, https://doi.org/10.1175/1520-0469(1963)020<0029:TGBOSC>2.0.CO;2, 1963.

Koptev, A. P., and Voskresenskii, A.I.: On the radiation properties of clouds. *Proc. Arctic and Antarctic Res. Inst.,* **239**, 39-47, 1962.

Korolev, A.: In-situ observation of Arctic mixed phase clouds during the ISDAC flight campaign (2010 - 13CldPhy13AtRad_13cldphy), MOCA Joint Symposium 09, Montreal, QC, 2010.

Korolev, A. and Leisner, T.: Review of experimental studies of secondary ice production, Atmospheric Chemistry and
Physics, 20, 11767–11797, https://doi.org/10.5194/acp-20-11767-2020, 2020.

Korolev, A. V., Emery, E. F., Strapp, J. W., Cober, S. G., Isaac, G. A., Wasey, M., and Marcotte, D.: Small Ice Particles in Tropospheric Clouds: Fact or Artifact? Airborne Icing Instrumentation Evaluation Experiment, Bulletin of the American Meteorological Society, 92, 967–973, https://doi.org/10.1175/2010BAMS3141.1, 2011.

Korolev, A., Heckman, I., Wolde, M., Ackerman, A. S., Fridlind, A. M., Ladino, L. A., Lawson, R. P., Milbrandt, J.,
and Williams, E.: A new look at the environmental conditions favorable to secondary ice production, Atmos. Chem. Phys., 20, 1391–1429, https://doi.org/10.5194/acp-20-1391-2020, 2020.

Korolev, A., Qu, Z., Milbrandt, J., Heckman, I., Cholette, M., Wolde, M., Nguyen, C., McFarquhar, G. M., Lawson, P., and Fridlind, A. M.: High ice water content in tropical mesoscale convective systems (a conceptual model), Atmospheric Chemistry and Physics, 24, 11849–11881, https://doi.org/10.5194/acp-24-11849-2024, 2024.

Korolev, A. V., Strapp, J. W., Isaac, G. A., and Nevzorov, A. N.: The Nevzorov Airborne Hot-Wire LWC–TWC Probe: Principle of Operation and Performance Characteristics, J. Atmos. Oceanic Technol., 15, 1495–1510, https://doi.org/10.1175/1520-0426(1998)015<1495:TNAHWL>2.0.CO;2, 1998.

Lawson, R. P. and Cooper, W. A.: Performance of Some Airborne Thermometers in Clouds, J. Atmos. Oceanic Technol., 7, 480–494, https://doi.org/10.1175/1520-0426(1990)007<0480:POSATI>2.0.CO;2, 1990.

Lawson, R. P., Baker, B. A., Schmitt, C. G., and Jensen, T. L.: An overview of microphysical properties of Arctic clouds observed in May and July 1998 during FIRE ACE, J. Geophys. Res., 106, 14989–15014, https://doi.org/10.1029/2000JD900789, 2001.

Lawson, R. P., and. Baker, B. A.: Improvement in determination of ice water content from two-dimensional particle imagery. Part II: Applications to collected data. Journal of Applied Meteorology, 45, 1291-1303, 2006
Lawson, P., Gurganus, C., Woods, S., and Bruintjes, R.: Aircraft Observations of Cumulus Microphysics Ranging from the Tropics to Midlatitudes: Implications for a "New" Secondary Ice Process, Journal of the Atmospheric Sciences, 74, 2899–2920, https://doi.org/10.1175/JAS-D-17-0033.1, 2017.

Lawson, R. P.: Effects of ice particles shattering on the 2D-S probe, Atmospheric Measurement Techniques, 4, 1361–1381, https://doi.org/10.5194/amt-4-1361-2011, 2011.

Lawson, R. P. and Zuidema, P.: Aircraft Microphysical and Surface-Based Radar Observations of Summertime Arctic Clouds, Journal of the Atmospheric Sciences, 66, 3505–3529, https://doi.org/10.1175/2009JAS3177.1, 2009.

Lawson, R. P., Stamnes, K., Stamnes, J., Zmarzly, P., Koskuliks, J., Roden, C., Mo, Q., Carrithers, M., and Bland, G. L.: Deployment of a Tethered-Balloon System for Microphysics and Radiative Measurements in Mixed-Phase Clouds at Ny-Ålesund and South Pole, Journal of Atmospheric and Oceanic Technology, 28, 656–670,
https://doi.org/10.1175/2010JTECHA1439.1, 2011.

Lawson, R. P., Woods, S., and Morrison, H.: The Microphysics of Ice and Precipitation Development in Tropical Cumulus Clouds, Journal of the Atmospheric Sciences, 72, 2429–2445, https://doi.org/10.1175/JAS-D-14-0274.1, 2015.

Lawson, R. P., Bruintjes, R., Woods, S., and Gurganus, C.: Coalescence and Secondary Ice Development in Cumulus Congestus Clouds, Journal of the Atmospheric Sciences, 79, 953–972, https://doi.org/10.1175/JAS-D-21-0188.1, 2022.

Lawson, R. P., Korolev, A. V., DeMott, P. J., Heymsfield, A. J., Bruintjes, R. T., Wolff, C. A., Woods, S., Patnaude, R. J., Jensen, J. B., Moore, K. A., Heckman, I., Rosky, E., Haggerty, J., Perkins, R. J., Fisher, T., and Hill, T. C. J.: The Secondary Production of Ice in Cumulus Experiment (SPICULE), Bulletin of the American Meteorological Society, 104, E51–E76, https://doi.org/10.1175/BAMS-D-21-0209.1, 2023.

Lilie, L., Emery, E., Strapp, J., and Emery, J.: A Multiwire Hot-Wire Device for Measurment of Icing Severity, Total Water Content, Liquid Water Content, and Droplet Diameter, in: 43rd AIAA Aerospace Sciences Meeting and Exhibit, 43rd AIAA Aerospace Sciences Meeting and Exhibit, Reno, Nevada, https://doi.org/10.2514/6.2005-859.

Luke, E. P., Yang, F., Kollias, P., Vogelmann, A. M., and M. Maahn, M.: New insights into ice multiplication using remote-sensing observations of slightly supercooled mixed-phase clouds in the Arctic, Proc. Natl. Acad. Sci.. 118, e2021387118, https://doi.org/10.1073/pnas.2021387118, 2021.

Marcolli, C.: Pre-activation of aerosol particles by ice preserved in pores, Atmos. Chem. Phys., 17, 1595–1622, https://doi.org/10.5194/acp-17-1595-2017, 2017.

McFarquhar, G. M., Ghan, S., Verlinde, J., Korolev, A., Strapp, J. W., Schmid, B., Tomlinson, J. M., Wolde, M., Brooks, S. D., Cziczo, D., Dubey, M. K., Fan, J., Flynn, C., Gultepe, I., Hubbe, J., Gilles, M. K., Laskin, A., Lawson, P., Leaitch, W. R., Liu, P., Liu, X., Lubin, D., Mazzoleni, C., Macdonald, A.-M., Moffet, R. C., Morrison, H., Ovchinnikov, M., Shupe, M. D., Turner, D. D., Xie, S., Zelenyuk, A., Bae, K., Freer, M., and Glen, A.: Indirect and Semi-direct Aerosol Campaign: The Impact of Arctic Aerosols on Clouds, Bull. Amer. Meteor. Soc., 92, 183–201, https://doi.org/10.1175/2010BAMS2935.1, 2011.

Mioche, G., Jourdan, O., Delanoë, J., Gourbeyre, C., Febvre, G., Dupuy, R., Monier, M., Szczap, F., Schwarzenboeck, A., and Gayet, J.-F.: Vertical distribution of microphysical properties of Arctic springtime low-level mixed-phase clouds over the Greenland and Norwegian seas, Atmospheric Chemistry and Physics, 17, 12845–12869, https://doi.org/10.5194/acp-17-12845-2017, 2017.

Mitchell, D. L.: Use of Mass- and Area-Dimensional Power Laws for Determining Precipitation Particle Terminal Velocities, J. Atmos. Sci., 53, 1710–1723, https://doi.org/10.1175/1520-0469(1996)053<1710:UOMAAD>2.0.CO;2, 1996.

Morrison, H., de Boer, G., Feingold, G., Jerry Harrington, J., Shupe, M., and K. Sulia,: Resilience of persistent Arctic mixed-phase clouds. Nature Geosci.Rev. 5, 11-17, https://doi.org/10.1038/ngeo1332, 2012.

Mossop, S. C.: Sublimation nuclei, P. Phys. Soc. B, 69, 161–164, doi:10.1088/0370-1301/69/2/305, 1956.

Mossop, S., Ruskin, R., & Heffernan, K.: Glaciation of a cumulus at approximately- 4c. Journal of the Atmospheric Sciences, 25(5), 889–899. https://doi.org/10.1175/1520-0469(1968)025<0889:goacaa>2.0.co;2, 1968.

Mossop, S. C.: Secondary ice particle production during rime growth: The effect of drop size distribution and rimer velocity, Quart J Royal Meteoro Soc, 111, 1113–1124, https://doi.org/10.1002/qj.49711147012, 1985.

Mossop, S. C. and Hallett, J.: Ice Crystal Concentration in Cumulus Clouds: Influence of the Drop Spectrum, Science, 186, 632–634, https://doi.org/10.1126/science.186.4164.632, 1974.

Pazmany, A. L. and Haimov, S. J.: Coherent Power Measurements with a Compact Airborne Ka-Band Precipitation

Radar, Journal of Atmospheric and Oceanic Technology, 35, 3–20, https://doi.org/10.1175/JTECH-D-17-0058.1, 2018.

Perkins, R.: Personal Communication, 2025.

Phillips, V. T. J., Patade, S., Gutierrez, J., and Bansemer, A.: Secondary Ice Production by Fragmentation of Freezing Drops: Formulation and Theory, Journal of the Atmospheric Sciences, 75, 3031–3070, https://doi.org/10.1175/JAS-D-17-0190.1, 2018.

Pruppacher, H. R. and Klett, J. D.: Microphysics of Clouds and Precipitation, Springer Science & Business Media, 975 pp., 2010.

Saunders, C. P. R. and Hosseini, A. S.: A laboratory study of the effect of velocity on Hallett–Mossop ice crystal multiplication, Atmospheric Research, 59, 3–14, https://doi.org/10.1016/S0169-8095(01)00106-5, 2001.

Seidel, J. S., Kiselev, A. A., Keinert, A., Stratmann, F., Leisner, T., and Hartmann, S.: Secondary ice production – no evidence of efficient rime-splintering mechanism, Atmospheric Chemistry and Physics, 24, 5247–5263, https://doi.org/10.5194/acp-24-5247-2024, 2024.

Shupe, M. D., Walden, V. P., Eloranta, E., Uttal, T., Campbell, J. R., Starkweather, S. M., and Shiobara, M.: Clouds at Arctic Atmospheric Observatories, Part I: Occurrence and macrophysical properties, J. Appl. Meteor. Clim., 50, 626–644, 2011.

Shupe, M. D., Persson, P. O. G., Brooks, I. M., Tjernström, M., Sedlar, J., Mauritsen, T., Sjogren, S., and Leck, C.: Cloud and boundary layer interactions over the Arctic sea ice in late summer, Atmos. Chem. Phys., 13, 9379–9399,
doi:10.5194/acp-13-9379-2013, 2013.

Shupe, M. D., Kollias, P., Persson, P. O. G., and McFarquhar, G. M.: Vertical Motions in Arctic Mixed-Phase Stratiform Clouds, Journal of the Atmospheric Sciences, 65, 1304–1322, https://doi.org/10.1175/2007JAS2479.1, 2008.

Solomon, A., Shupe, M. D., Persson, P. O. G., and Morrison, H.: Moisture and dynamical interactions maintaining
decoupled Arctic mixed-phase stratocumulus in the presence of a humidity inversion, Atmospheric Chemistry and Physics, 11, 10127–10148, https://doi.org/10.5194/acp-11-10127-2011, 2011.

Solomon, A., Feingold, G., and Shupe, M. D.: The role of ice nuclei recycling in the maintenance of cloud ice in Arctic mixed-phase stratocumulus, Atmospheric Chemistry and Physics, 15, 10631–10643, https://doi.org/10.5194/acp-15-10631-2015, 2015.

Verlinde, J., Harrington, J. Y., McFarquhar, G. M., Yannuzzi, V. T., Avramov, A., Greenberg, S., Johnson, N., Zhang, G., Poellot, M. R., Mather, J. H., Turner, D. D., Eloranta, E. W., Zak, B. D., Prenni, A. J., Daniel, J. S., Kok, G. L., Tobin, D. C., Holz, R., Sassen, K., Spangenberg, D., Minnis, P., Tooman, T. P., Ivey, M. D., Richardson, S. J., Bahrmann, C. P., Shupe, M., DeMott, P. J., Heymsfield, A. J., and Schofield, R.: The Mixed-Phase Arctic Cloud Experiment, Bull. Amer. Meteor. Soc., 88, 205–222, https://doi.org/10.1175/BAMS-88-2-205, 2007.

Wang, Z.: Personal Communication, 2025.

Wang, Z., et al., Multi-functional Airborne Raman Lidar (MARLi) and 5-Beam Airborne Doppler Radar (ADL).Available at: https://www.eol.ucar.edu/sites/default/files/2023-12/WANG_MARLi_ADL_Wang.pdf, 2023.

Wendisch, M., Macke, A., Ehrlich, A., Lüpkes, C., Mech, M., Chechin, D., Dethloff, K., Velasco, C. B., Bozem, H., Brückner, M., Clemen, H.-C., Crewell, S., Donth, T., Dupuy, R., Ebell, K., Egerer, U., Engelmann, R., Engler, C.,
Eppers, O., Gehrmann, M., Gong, X., Gottschalk, M., Gourbeyre, C., Griesche, H., Hartmann, J., Hartmann, M., Heinold, B., Herber, A., Herrmann, H., Heygster, G., Hoor, P., Jafariserajehlou, S., Jäkel, E., Järvinen, E., Jourdan, O., Kästner, U., Kecorius, S., Knudsen, E. M., Köllner, F., Kretzschmar, J., Lelli, L., Leroy, D., Maturilli, M., Mei, L., Mertes, S., Mioche, G., Neuber, R., Nicolaus, M., Nomokonova, T., Notholt, J., Palm, M., Van Pinxteren, M.,

Quaas, J., Richter, P., Ruiz-Donoso, E., Schäfer, M., Schmieder, K., Schnaiter, M., Schneider, J., Schwarzenböck, A., Seifert, P., Shupe, M. D., Siebert, H., Spreen, G., Stapf, J., Stratmann, F., Vogl, T., Welti, A., Wex, H., Wiedensohler, A., Zanatta, M., and Zeppenfeld, S.: The Arctic Cloud Puzzle: Using ACLOUD/PASCAL Multiplatform Observations to Unravel the Role of Clouds and Aerosol Particles in Arctic Amplification, Bulletin of the American Meteorological Society, 100, 841–871, https://doi.org/10.1175/BAMS-D-18-0072.1, 2019.

Yang, F., Ovchinnikov, M., and Shaw, R. A.: Long-lifetime ice particles in mixed-phase stratiform clouds: Quasi-steady and recycled growth, JGR Atmospheres, 120, https://doi.org/10.1002/2015JD023679, 2015.

Yang J., Z. Wang, Z., Heymsfield, A., DeMott P. J., Twohy, C. H., Suski, K. J. and Toohey, D. W.: High ice, concentration observed in tropical maritime stratifrom mixed-phase clouds with top temperature warmer than -8C, Atmospheric Research, 233, https://doi.org/10.1016/j.atmosres.2019.104719, 2020.


