# Peer review of "Micrometeorology of Arctic Stratiform Boundary-layer Clouds during ARCSIX"

_EGUsphere, 2025_

## Author Comment (AC1)

We thank Prof. French for his insightful and helpful comments. We are especially grateful for his acknowledgement that it is not always necessary to solve all the questions raised by the observations. Our replies are in red text and new line numbers in the revised manuscript are in red text. We apologize for the clutter in the Track Changes version. Due to new figures and multiple changes in the text it is difficult to follow.

The authors present measurements of cloud microphysical structure from several aircraft-observed cases of stratiform boundary-layer clouds (SBCs) during the recent (mid-2024) ARCSIX campaign. The manuscript focuses on cases with relatively warm cloud tops (T > -9 deg C and T=>-4 deg C in extreme cases) that contain ice, either widespread or in isolated pockets. The authors assert several times that there are no good explanations, based on our current understanding of ice nucleation, to explain the relatively high ice concentrations that were observed in some of these clouds. The authors present a very well written and thorough discussion providing some conjecture of how ice may have been initiated in these clouds. In the end, the authors admit that the observations along with their interpretation provide more questions than answers. I agree with their assessment.

I found the manuscript well-written and quite enjoyable to read. While I might disagree on a few minor points in the paper, I think this has more to do with style than actual substance. I do find it refreshing to read a paper that isn't able to 'solve' all of the questions raised by the observations and to admit that there are some aspects of cloud evolution, especially in mixed-phase conditions, that we do not fully understand. The authors do a good job of pointing back to previous measurements in the arctic to demonstrate that others have made similar measurements. This provides confidence in the measurements provided here and demonstrates this isn't a 'new' problem, but it is a timely one!

I recommend accepting with minor changes.

**Broad/Major comment:**

1. INP measurements quoted in the paper come both from ARCSIX and MOSAIC and are based on filter measurements. From ARCSIX the measurements are quite limited because of the low collection time (length of time filters are exposed), but this is less of a problem with MOSAIC. Regardless, this methodology of collecting filters and processing the samples offline requires assumptions about mode of nucleation and how particles interact with hydrometeors, etc. I am *not* asserting these assumptions are wrong, but I do wonder out loud if the methodology does not lead to limitations of describing what primary nucleation might be able to produce in conditions that do not adhere strictly to those associated with determining INPs from filter samples. I bring this point up not because I necessarily disagree with the author's assessment and/or conjecture in the paper, but rather because I think the lack of explanation for the observations based on our current scientific understanding can provide even more opportunity for the author's to be a little more provocative in exploring ways to explain the measurements.

Response: We appreciate the reviewer's concerns regarding the offline processing of INPs from filters, and how implicit assumptions about the mode of nucleation and particle interaction

accurately represents primary nucleation in natural clouds. We have added a paragraph in the summary (new ms lines 607 – 615) of the revised manuscript further describing limitations of offline processing of filters, and how these conditions do not strictly represent primary nucleation in natural clouds.

"As cogently pointed out by a reviewer, the current technology for measuring INPs as a function of temperature may not be adequate to explain how primary nucleation actually takes place in natural clouds. Collection of aerosols on filters followed by offline immersion freezing obviously introduces artificial factors that do not exist in natural clouds. Airborne INP processing using a Continuous Flow Diffusion Chamber (CFDC) attempts to simulate cloud conditions, but cannot accurately simulate repeated cycling of INPs or time dependence. It may be possible for large cloud chambers to simulate repeated cycling of INPs, but this has not been attempted at $T \geq$ -4 °C. The above discussion is not intended to be a slam on current INP measurement methodologies. The purpose of this critique is to point out that in-situ cloud particle measurements and laboratory simulations cannot possibly encompass the complex physical and chemical interactions that impact primary nucleation in natural clouds."

**Minor comments:**

1. Line 33, abstract: "The *total* length of clouds…" delete total here, it is redundant.

Deleted.

2. Line 42, abstract: change *analyzes* to *analysis*

Changed.

3. Line 79 & 80: Morrison et al. (2012) is missing in references; I'm not sure what is meant by "the predominance of Arctic mixed-phase clouds is largely due to their longevity." (I was going to dig up the reference to see if I could figure it out from there…) Are the authors stating that if arctic clouds were not long-lived they would be mostly all liquid or all ice? That doesn't seem correct?

Response: We apologize for the missing reference and confusing sentence. We have added the Morrison et al., (2012) reference and modified the sentence to use language found in the Morrison paper.

Lines 78 – 80: "In a review article, Morrison et al., (2012) cites long-term, ground-based observations from Shupe et al., (2011) showing that mixed-phase clouds cover large swaths of the Arctic throughout the year. Morrison et al., (2012) further explains that the high frequency of occurrence of Arctic mixed-phase clouds is largely owing to their longevity."

4. Line 81 and 82: I suggest changing the sentence slightly to state that there exists several reports of observations of mixed clouds containing concentrations of ice crystals greater than what would be expected from primary nucleation at cloud top temperatures (…or the coldest cloud temperature).

Done

5. Line 86: …temperature of -6…*add the word 'of'*

Done

6. Lines 119-121: clarify that Shupe et al. (2011) note that *when boundary layer clouds clouds are present,* they are mixed phase 60% of the time and all liquid 40% of the time.

Reworded as suggested

7. Figure 2: in fig 2b, one of the Learjet tracks is missing (the one directly north of Pituffik around 80 deg latitude).

Good Catch – corrected.

8. Line 140-1: "…and produces gray images at more than 20 times the data rate of previous gray probes." I don't understand this statement. Is this asserting that other gray probes 'miss' 19 out of 20 particles? Certainly the 2DS has higher resolution than other gray-scale probes such as the 25 micron CIP-gray, but that would lead to 2.5 times better resolution for the 2DS (and hence data rate), not 20 times? Data rate seems an odd measure here…

Yes, this could be expressed as the older (CIP gray) probes missing 19 out of 20 particles. The reason is attributable to the electronics used to process the images. The CIP gray probe reads out each gray level in serial. The SPEC probe reads out all of the gray levels in parallel, plus the electronics in the SPEC probe is updated to process the image data considerably faster. This results in "approximately" a 20 times greater data rate, i.e., 20 times as many particle images are recorded. Since SPEC has not performed a side-by-side comparison with the CIP  gray probe, which would produce a more accurate comparison, it may be more appropriate to reword the sentence to read: "more than an order of magnitude greater than…"

9. Lines 148-159: It would be useful to add one or two sentences to describe how the KPR data are processed/presented here. Are the authors using the 'native' resolution of the radar, and if so what is it? Are they smoothing the radar measurements along track? Along range?

Lines 158 – 166: We have added more detail describing how the radar measurements were collected and how the data were processed.

10. Line 165: the authors quote drop(let) concentrations averaged over P3 and Learjet measurements of 65+/-59 cm-3. I don't understand what this measurement represents? Is this all of the incloud measurements averaged together (over all flights)? If so, is the +/-59 then represent 1 standard deviation? One quartile? Or does this measurement represent something else altogether?

We have now added new Figs. 4 and 5 that show probability distribution plots of the measurements and the mean concentration of ice as a function of temperature. There is also discussion of these figures in lines 168 - 206

11. In figure 4, and in figure 8, and in figure 10 – in the mixed phase regions, based on the LWC/TWC measurements the ice water content is basically zero…that is not all that surprising that in these regions the vast majority of the mass is being carried by liquid substance…however, I do wonder if these estimates (or my interpretation of them) is consistent to what would be predicted of IWC from the OAP measurements? Also, I think the authors should note in the description of these cases that although in the mixed phase region, the ice particles concentration is high, the amount of mass carried in the ice phase is still incredibly small compared to the total condensed mass.

The reviewer brings out a good point. We have added the average IWC in Figs. 6, 10 and 12 (new figure numbers), and inserted text before new Fig. 6 that makes the reviewer's point. A comparison of LWC and IWC can also be seen in new Fig. 4.

12. Line 240 (in reference to figure 6)…the correlation noted in the text between ice concentration, IWC, extinction and lidar measurements are NOT with the lidar surface return, but rather the lidar return *above* the surface. I found the labeling of 'lidar surface return' confusing since we are not interested in the surface return, but rather return between the aircraft flight level and surface.

New Fig. 8: This was labeled incorrectly.  LSR is an acronym for both Lidar Surface Return and Lidar Scattering Ratio. We discussed the figure with Zhien Wang, who looked further into the MARLi measurements. He explained that there is a surface return that varies with surface properties (e.g., ice or open water), but in this case the lidar is seeing the high concentrations of precipitating ice. He noted that values of Lidar Scattering Ratio derived from lidar measurements also peaked at values of about 4 km$^{-1}$.

13. Line 315: replace 'ignored completely' with 'dismissed'.

Done

14. Lines 351-2, and 365-6: Why do the authors assert here that all ice crystals result from primary nucleation? I understand there is no apparent mechanism to explain SIP, but then again there is no good understanding to support primary? Since it cannot be explained, I'm not sure it can be assigned all to primary?

Lines 383 – 394: We agree that there is no evidence to support "primary" nucleation. To avoid this unsupported assumption, we have deleted "primary" in this discussion. At the recommendation of another reviewer, we also added a sentence noting that the presence of drizzle increases the probability of immersion freezing due to the volume effect.

15. Lines 360-4: I **think** the authors are arguing low concentration of graupel below cloud is because the graupel is not growing via collisions due to its nearly identical fall velocity of drizzle drops. But the graupel would also grow via deposition and as it fell would collect

drizzle, so I'm not sure I follow the argument. I don't necessarily have a better one, though….

The graupel would grow by deposition, but not be colliding with the drizzle since they are falling at nearly the same terminal velocity and turbulence is very low in these clouds (albeit, turbulence measurements are not shown and we do not want to open that can of worms). However, the linear depositional growth of quasi-spherical graupel will be much smaller than the linear growth of columnar ice, which grows by deposition mainly along its c-axis.

16. Line 374-5: The authors bring up a great point!!!!

Thanks!

17. Figure 9e: The reflectivity labels on the dBZ colorscale needs to be darker, it is not readable.

The figure has been redone as recommended.

18. Line 438: 'measurements of temperature, altitude, microphysical measurements and…' change the second 'measurements' to 'parameters' (or 'characteristics')

Done

19. Line 488-9: the sentence beginning 'Of the total….' Needs to be re-worded somehow.

Reworded with an attempt to make this sentence clearer.

20. Line 543: change 'like' to likely

Done

---

## Author Comment (AC2)

We thank the reviewer for his careful analysis of our paper and thoughtful comments. Our replies are in red text and new line numbers in the revised manuscript are in red text. We apologize for the clutter in the Track Changes version. Due to new figures and multiple changes in the text it is difficult to follow.

This manuscript presents a highly valuable and comprehensive dataset of in-situ and remote-sensing observations of stratiform boundary-layer clouds collected during the 2024 ARCSIX campaign north of Greenland. Using two research aircraft (NASA P-3 and the SPEC Inc. Learjet) equipped with state-of-the-art microphysical probes—and additionally, aerosol and radar instrumentation—the ARCSIX campaign sampled more than 12,000 km of clouds, including 6,266 km of in-situ measurements. The resulting dataset, covering spring and summer conditions, represents one of the most extensive contemporary observational efforts targeting Arctic mixed-phase clouds.

The study addresses one of the most significant unresolved topics in cloud microphysics: secondary ice production (SIP) in mixed-phase clouds. SIP is known to dominate ice crystal concentrations under certain environmental conditions, yet these conditions—along with the relative contributions of different SIP mechanisms—remain poorly constrained. In this context, the manuscript provides an important observational framework for understanding under which thermodynamic and microphysical conditions SIP may occur in Arctic low-level clouds.

A key strength of the work is the combination of in-situ sampling with remote sensing (Ka-band radar). This approach is especially valuable because in-situ measurements, while precise, offer only localized and momentary snapshots, whereas remote sensing helps place them into a broader spatial and temporal perspective.

The authors document several cases of anomalously high ice concentrations at very warm cloud-top temperatures ($\geq -4\ °C$), far exceeding what can be explained by measured INP concentrations. Importantly, the authors excluded seeding from above and lofting from the surface (including blown snow), strengthening the case that the observed ice is produced internally. While clouds frequently met widely accepted temperature and microphysical conditions for known SIP mechanisms such as Hallett–Mossop (HM), these mechanisms cannot explain all cases—particularly the very warm mixed-phase clouds discussed in the study. Conversely, some clouds that fulfilled HM criteria did not exhibit enhanced ice, emphasizing that the environmental controls on SIP remain elusive.

The manuscript further offers insightful hypotheses regarding additional factors that should be considered in SIP theory—most notably, the size and concentration of graupel, which may play a critical role in the onset or efficiency of HM SIP. The discussion of frozen-drop fragmentation (FFD) is balanced and appropriately cautious given the suboptimal environmental conditions for this mechanism in ARCSIX clouds.

Overall, the manuscript is well written, enjoyable to read, and provides important observations that challenge current understanding of both primary and secondary ice formation in Arctic mixed-phase clouds. The ARCSIX dataset will undoubtedly serve as a cornerstone for future laboratory, modeling, and observational work aimed at disentangling the still-poorly-understood

conditions that trigger SIP. Therefore, I recommend the manuscript for publication with some minor suggestions.

**Minor suggestions:**

**1**. The ARCSIX campaign sampled a broad range of cloud conditions. While the authors provide a verbal summary of cloud-phase occurrence and include a comprehensive flight table in the supplementary material, an additional overview figure summarizing the campaign-wide findings would help place the selected case studies into a broader observational context. Since the focus of the paper is on ice production, a figure showing, for example, ice crystal number concentration or ice water content as a function of temperature (or a similar integrative metric) would be particularly useful.

We agree. Please see newly-added Figs. 4 and 5 and accompanying text.

**2**. The manuscript frequently refers to the presence (or absence) of graupel as a key criterion for Hallett–Mossop (HM) rime splintering, and in some cases questions the applicability of HM SIP when insufficient graupel is identified. However, in the classical description of the HM process, the essential requirement is the presence of riming particles, not graupel per se. To my knowledge, any sufficiently large ice particle (typically D ≳ 300 μm) collecting supercooled droplets can act as a rimer and potentially generate splinters (e.g., Hallett & Mossop, 1974; Mossop, 1980). In this context, the observed larger columnar or needle-shaped ice crystals could plausibly serve as rimers. It may therefore be more appropriate to frame the discussion in terms of rimers rather than graupel, which would avoid unnecessarily restrictive assumptions about particle habit and better align with the original HM framework.

We have addressed the reviewer's concern in detail in the third and fourth paragraphs of Section 4.2.1.

**Lines 98-101**: The authors mention that relatively high ice crystal concentrations in stratiform clouds are not unique to the Arctic. This is true, but it would be advised to add as examples studies from Southern Ocean, as also one of the first reported observations of SIP was done in this region (e.g., Mossop et al., 1968).

The reference has been added – Thank you.

**Lines 173-175**: Can you give a size range for the ice particle concentrations?

See newly inserted figures 4 and 5.

**Lines 226-227**: How does the statement that irregular ice crystals are not expected at cloud top temperatures of -7°C fit to the earlier studies that report <1 % pristine fractions in Arctic mixed-phase clouds (Korolev et al., 1999)?

There is no significant relationship to the ARCSIX boundary-layer clouds with $T \geq -7$ °C because data in the Korolev et al., (1999) paper were collected in cirrus and stratus clouds associated with frontal systems, and the large majority of the measurements were in clouds colder than -7 °C

**Lines 351-352 and 365-366**: The authors talk about primary ice nucleation at -6.1°C but primary ice nucleation alone cannot explain the observed ice crystal concentrations.

Lines 383 – 391: We agree with the reviewer. "Primary" has been deleted.

**356-357**: Given the large volumes and long residence times of drizzle-sized droplets, to what extent can freezing at relatively warm temperatures be explained by size-dependent freezing probabilities rather than invoking extremely rare ice-nucleating particles?

Lines 392 – 394: This is a good point. We added a sentence noting the increased probability of immersion freezing as a function of drop volume.

**Figure 10**: Including vertical wind velocity in the time series, if available, could help to assess the potential role of cellular dynamics.

We would like to show meaningful vertical velocity measurements, but unfortunately the variance in vertical velocity is typically very small, about $\pm 1$ m s$^{-1}$, which is on the order of the precision of the instrument.

**References**

Hallett, J., & Mossop, S. (1974). Production of secondary ice particles during the riming process. Nature, 249(5452), 26–28. https://doi. org/10.1038/249026a0

Korolev, A. V.; Isaac, G. A.; Hallett, J. Ice Particle Habits in Arctic Clouds. Geophysical Research Letters 1999, 26 (9), 1299–1302. https://doi.org/10.1029/1999GL900232.

Mossop, S., Ruskin, R., & Heffernan, K. (1968). Glaciation of a cumulus at approximately- 4c. Journal of the Atmospheric Sciences, 25(5), 889–899. https://doi.org/10.1175/1520-0469(1968)025<0889:goacaa>2.0.co;2

Mossop, S. (1980). The mechanism of ice splinter production during riming. Geophysical Research Letters, 7(2), 167–169. https://doi. org/10.1029/gl007i002p00167